# Fine-scale computations for adaptive processing in the human brain

Elisa Zamboni[1], Valentin G Kemper[2,3], Nuno Reis Goncalves[1], Ke Jia[1], Vasilis M Karlaftis[1], Samuel J Bell[1], Joseph Giorgio[1], Reuben Rideaux[1], Rainer Goebel[2,3], Zoe Kourtzi[1]*

[1]Department of Psychology, University of Cambridge, Cambridge, United Kingdom; [2]Department of Cognitive Neuroscience, Faculty of Psychology and Neuroscience, Maastricht University, Maastricht, Netherlands; [3]Department of Cognitive Neuroscience, Maastricht Brain Imaging Center, Maastricht University, Maastricht, Netherlands

**Abstract** Adapting to the environment statistics by reducing brain responses to repetitive sensory information is key for efficient information processing. Yet, the fine-scale computations that support this adaptive processing in the human brain remain largely unknown. Here, we capitalise on the sub-millimetre resolution of ultra-high field imaging to examine functional magnetic resonance imaging signals across cortical depth and discern competing hypotheses about the brain mechanisms (feedforward vs. feedback) that mediate adaptive processing. We demonstrate layer-specific suppressive processing within visual cortex, as indicated by stronger BOLD decrease in superficial and middle than deeper layers for gratings that were repeatedly presented at the same orientation. Further, we show altered functional connectivity for adaptation: enhanced feedforward connectivity from V1 to higher visual areas, short-range feedback connectivity between V1 and V2, and long-range feedback occipito-parietal connectivity. Our findings provide evidence for a circuit of local recurrent and feedback interactions that mediate rapid brain plasticity for adaptive information processing.

*For correspondence:
zk240@cam.ac.uk

## Introduction

Interacting in cluttered and complex environments, we are bombarded with plethora of sensory information from diverse sources. The brain is known to address this challenge by reducing its responses to repeatedly or continuously presented sensory inputs (for reviews: *Clifford, 2002*; *Kohn, 2007*). This type of sensory adaptation is a rapid form of plasticity that is critical for efficient processing and has been shown to involve changes in perceptual sensitivity (for review: *Clifford, 2002*) and neural selectivity (for review: *Kohn, 2007*). Numerous neurophysiological studies (for review: *Kohn, 2007*) have shown sensory adaptation to be associated with reduction in neuronal responses that are specific to the features of the adaptor. Functional brain imaging studies in humans have shown functional magnetic resonance imaging (fMRI) adaptation for low-level visual features (e.g. contrast, orientation, motion; for review *Larsson et al., 2016*) as indicated by decreased BOLD responses in visual cortex due to stimulus repetition. Similar BOLD decreases have been reported in higher visual areas for repeated presentation of more complex visual stimuli (e.g. faces objects), an effect known as repetition suppression (*Grill-Spector et al., 2006*; *Krekelberg et al., 2006*). Yet, the fine-scale human brain computations that underlie adaptive processing remain debated.

In particular, neurophysiological studies focussing on primary visual cortex provide evidence of rapid adaptation at early stages of sensory processing (*Gutnisky and Dragoi, 2008*; *Whitmire and Stanley, 2016*; *Xiang and Brown, 1998*). In contrast, fMRI studies have suggested top-down

influences on sensory processing of repeated stimuli via feedback mechanisms (e.g. *Ewbank et al., 2011*; *Summerfield et al., 2008*). Yet, the circuit mechanisms that mediate adaptive processing in the human brain remain largely unknown, as fMRI at standard resolution does not allow us to discern feedforward from feedback signals.

Here, we capitalise on recent advances in brain imaging technology to determine the contribution of feedforward vs. feedback mechanisms to adaptive processing. Ultra-high field (UHF) imaging affords the sub-millimetre resolution necessary to examine fMRI signals across cortical depth in a non-invasive manner, providing a unique approach to interrogate human brain circuits at a finer scale (for review: *Lawrence et al., 2019a*) than that possible by standard fMRI techniques (for review: *Goense et al., 2016*). UHF laminar imaging allows us to test the finer functional connectivity across cortical depth based on known anatomical laminar circuits. In particular, sensory inputs are known to enter the cortex from the thalamus at the level of the middle layer (layer 4), while output information is fed forward from superficial layers (layer 2/3), and feedback information is exchanged primarily between deeper layers (layer 5/6) as well as superficial layers (for review: *Self et al., 2019*).

Here, we combine UHF laminar imaging with an orientation adaptation paradigm (i.e. observers are presented with gratings at the same or different orientations) to test whether orientation-specific adaptation alters input processing in middle layers of visual areas (V1 and extrastriate visual areas), or feedback processing in superficial and deeper layers (*Figure 1A*). We demonstrate that adaptation alters orientation-specific signals across cortical depth in the visual cortex with stronger fMRI-adaptation (i.e. BOLD decrease for repeated stimuli) in superficial layers. This layer-specific fMRI adaptation relates to a perceptual bias in orientation discrimination due to adaptation, as measured using a tilt aftereffect paradigm (e.g. *Gibson and Radner, 1937*). Further, functional connectivity analysis shows that adaptation involves: (a) enhanced feedforward connectivity between V1 superficial layers and middle layers of extrastriate visual areas, and (b) enhanced short-range feedback between V2 and V1 deeper cortical layers. Finally, we test the role of the posterior parietal cortex in adaptive processing, as it is known to be involved in stimulus expectation and novelty detection (*de Lange et al., 2018*; *Garrido et al., 2009*; *Li et al., 2010*; *Summerfield and de Lange, 2014*). Our results show enhanced feedback connectivity from posterior parietal cortex to V1 deeper layers, suggesting top-down influences on visual processing via long-range feedback mechanisms. Our findings provide evidence for a circuit of local recurrent (i.e. feedforward and short-range feedback) processing across cortical depth in visual cortex, and occipito-parietal feedback interactions that mediate adaptive processing in the human brain.

## Results

### fMRI adaptation across cortical depth in visual cortex

To test whether adaptation alters visual orientation processing, we measured fMRI responses when participants (N = 15) were presented with gratings (N = 16 per block) either at the same orientation (adaptation) or different orientations (non-adaptation) in a blocked fMRI design (*Figure 1B*). Participants were asked to perform a Rapid Serial Visual Presentation (RSVP) task (i.e. detect a target in a stream of letters presented in the centre of the screen) to ensure that they attended similarly across conditions (*Larsson et al., 2006*).

We tested for fMRI adaptation in visual cortex due to stimulus repetition by comparing fMRI responses for adaptation (i.e. the same oriented sinewave grating presented repeatedly within a block) vs. non-adaptation (i.e. gratings of varying orientation presented in a block). To test for differences in orientation-specific fMRI adaptation across cortical depth, for each participant we mapped the retinotopic areas in the visual cortex (V1, V2, V3, V4), assigned voxels in three cortical depths (deeper, middle, superficial layers) using an equi-volume approach (see *Methods, MRI data analysis: Segmentation and cortical depth sampling*; *Figure 2D*), and extracted fMRI responses across cortical depths. To control for possible differences in thermal noise, physiological noise, or signal gain across cortical depths (*Goense et al., 2012*; *Havlicek and Uludağ, 2020*), we (a) matched the number of voxels across cortical depths for each participant and region of interest (ROI), and (b) z-scored the laminar-specific time courses to control for differences in variance across cortical depths, while preserving condition-dependent differences within each cortical layer.

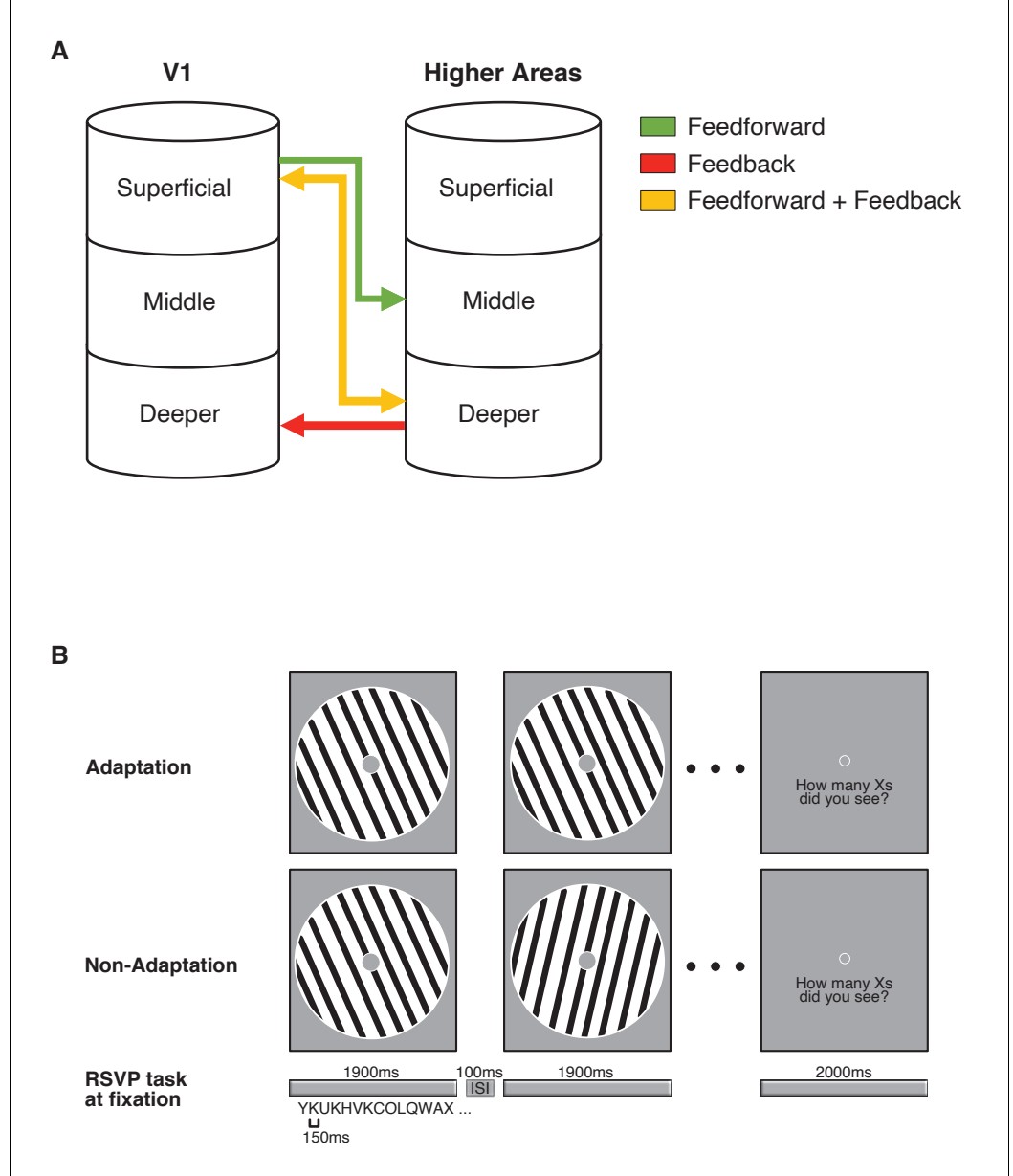

**Figure 1.** F unctional magnetic resonance imaging (fMRI) laminar circuits and fMRI design. (A) Schematic representation of feedforward (superficial – middle layers; green), feedback (deeper – deeper layers; red), and feedforward plus feedback (superficial – deeper layers; yellow) anatomical connectivity between V1 and higher cortical regions. Here, we focussed on feedforward vs. feedback connections. (B) fMRI design. Adaptation blocks comprised 16 sinewave gratings presented at the same orientation. Non-adaptation blocks comprised 16 gratings presented at different orientations. During stimulus presentation (1900 ms stimulus on, 100 ms stimulus off), participants were asked to perform an Rapid Serial Visual Presentation (RSVP) task; that is, count the number of times a target letter (e.g. X) was displayed in the stream of distracters and report it at the end of each stimulus block. Each letter was displayed for 150 ms and participants had 2000 ms to give their response.

Our results showed laminar-specific fMRI adaptation (i.e. decreased fMRI responses for adaptation compared to non-adaptation) in visual areas (*Figure 3A*). In particular, a repeated measures ANOVA showed significant main effects of ROI (V1, V2, V3, V4; $F_{(3,39)}=23.276$, $p=0.001$), cortical depth (deeper, middle, superficial; $F_{(2,26)}=4.942$, $p=0.034$), and condition (adaptation non-adaptation; $F_{(1,13)}=11.872$, $p=0.004$). There was no significant ROI × condition × cortical depth interaction ($F_{(6,78)}=1.949$, $p=0.131$), suggesting similar fMRI adaptation across visual areas. A significant

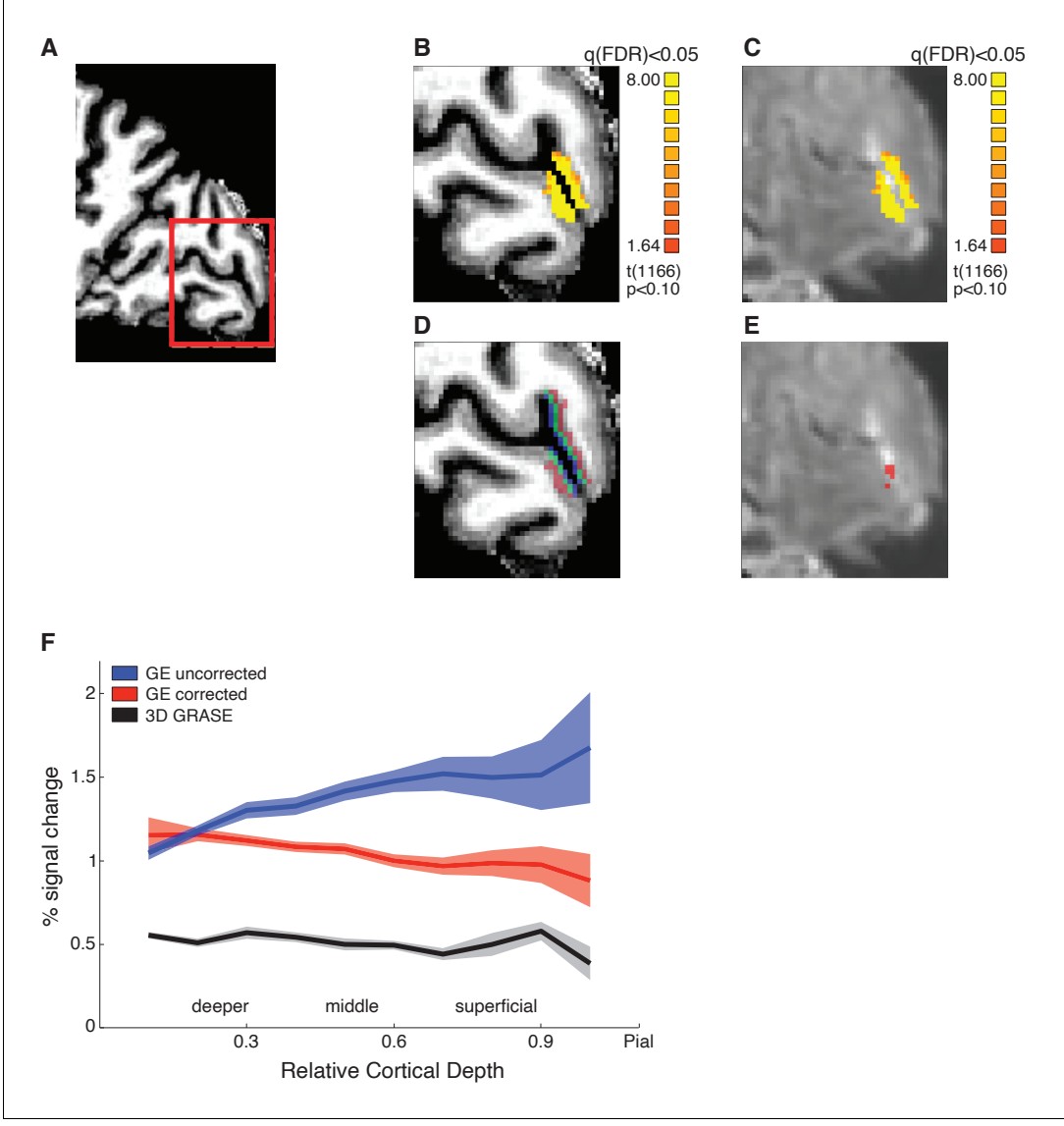

**Figure 2.** F unctional magnetic resonance imaging (fMRI) data analysis, layer segmentation, and vascular contribution correction. (**A**) Sagittal brain view of representative participant: red insert highlights region of interest (ROI, early visual cortex). Structural (**B**) and functional (**C**) images of the ROI showing activation maps for stimulus vs. fixation. Activation is well confined within the grey matter borders. (**D**) Mapping of cortical layers within the ROI: deeper layers shown in red, middle layers in green, superficial layers in blue. (**E**) Voxels confounded by vasculature effects (in red) overlaid on mean functional image. (**F**) Mean BOLD (per cent signal change from fixation baseline) across participants for V1 across cortical depth. Comparison between BOLD signal before (blue) and after temporal signal-to-noise ratio (tSNR) and t-value correction (red), and 3D GRASE BOLD signal (black). The superficial bias observed in the BOLD signal is reduced after correction and matches closely the laminar profile of the 3D GRASE data.

condition $\times$ cortical depth interaction ($F_{(2,26)}$=9.506, p=0.002) indicated stronger fMRI adaptation in superficial and middle than deeper layers. We observed a similar pattern of results when we controlled for signal contribution from voxels at the border of adjacent layers, using a spatial regression analysis (*Kok et al., 2016*; *Koster et al., 2018*; *Markuerkiaga et al., 2016*). To unmix the signal, we regressed out the time course of voxels assigned to middle layers and adjacent to the superficial layers from the time course of voxels assigned to superficial layers. We applied the same approach to voxels assigned to the deeper layers and adjacent to the middle layers. We observed stronger

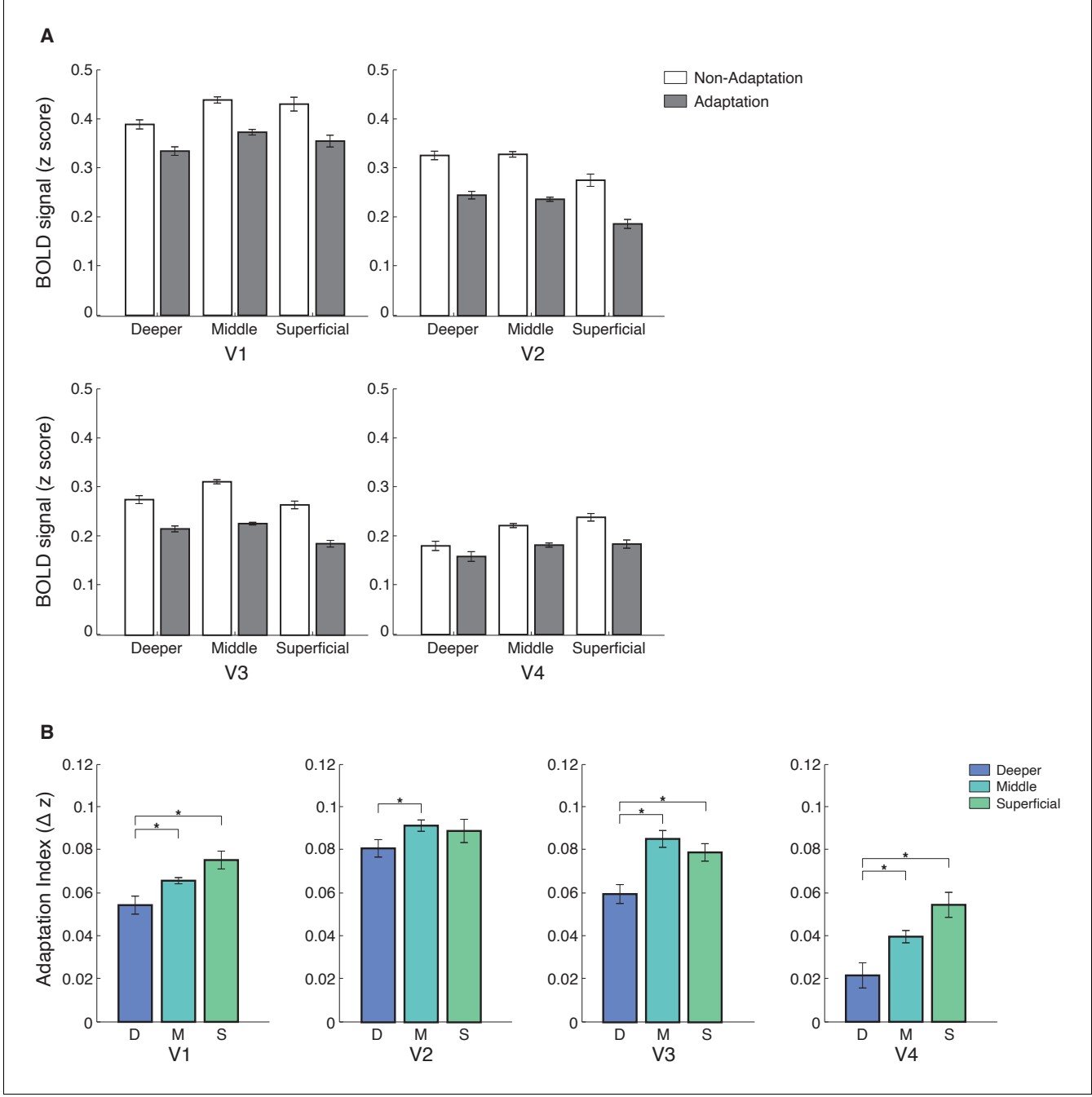

**Figure 3.** Laminar BOLD and fMRI adaptation index for V1, V2, V3, and V4. (**A**) Mean BOLD across V1, V2, V3, and V4 cortical layers. Bar-plot shows z-scored BOLD signal from fixation baseline for adaptation (grey) and non-adaptation (white) across cortical layers of V1, V2, V3, and V4. Error bars indicate within-subject confidence intervals (*Cousineau, 2005*) (N = 15 for V1, V2, and V3; N = 14 for V4). (**B**) fMRI adaptation index across cortical layers (D: deeper, M: middle, S: superficial) for V1, V2, V3, and V4. Bar-plots show difference in z-scored BOLD signal between the non-adaptation and the adaptation conditions. Error bars indicate within-subject confidence intervals (*Cousineau, 2005*) (N = 15 for V1, V2, and V3; N = 14 for V4). Stars indicate statistically significant comparisons for p<0.05.

The online version of this article includes the following source data and figure supplement(s) for figure 3:

**Source data 1.** Tables for mean BOLD responses to adaptation, non-adaptation, and adaptation index across cortical layers of areas V1, V2, V3, and V4.
**Figure supplement 1.** Behavioural tilt-aftereffect.
**Figure supplement 2.** 3D GRASE fMRI.

fMRI adaptation in superficial layers across visual areas following this correction (*Figure 3A*), as indicated by a significant condition × cortical depth interaction: F(2,26)=5.996, p=0.022.

To further quantify fMRI adaptation across cortical depths, we computed an fMRI adaptation index (i.e. fMRI responses for non-adaptation minus adaptation) per ROI and cortical depth (*Figure 3B*). A repeated-measures ANOVA showed a significant main effect of cortical depth (F (2,26)=9.506, p=0.002), ROI (F(3,39)=5.858, p=0.003), and no significant ROI × cortical depth interaction (F(6,78)=1.949, p=0.131). Post-hoc comparisons showed significantly stronger fMRI adaptation across visual areas in superficial compared to deeper layers (V1: t(14)=−2.556, p=0.023; V3: t (14)=−2.580, p=0.022; V4: t(13)=−2.091, p=0.012; with the exception of V2: t(14)=−0.881, p=0.393) and middle compared to deeper layers (V1: t(14)=−2.429, p=0.029; V2: t(14)=−2.524, p=0.024; V3: t(14)=−3.528, p=0.003; V4: t(13)=−2.519, p=0.026). No significant differences were observed between fMRI adaptation in superficial and middle cortical layers across visual areas (V1: t(14) =−2.093, p=0.055; V2: t(14)=0.331, p=0.746; V3: t(14)=0.942, p=0.362; V4: t(13)=−2.096, p=0.056).

## Control analyses

To control for potential confounds due to the contribution of vasculature-related signals to BOLD, we conducted the following additional analyses.

First, it is known that BOLD measured by GE-EPI is higher at the cortical surface due to vascular contributions (*Uğurbil et al., 2003*; *Uludağ et al., 2009*; *Yacoub et al., 2005*). To ensure that the fMRI adaptation we observed in superficial layers was not confounded by this superficial bias, we identified and removed voxels with low temporal signal-to-noise ratio (tSNR) and high t-statistic for stimulation contrast (see Materials and methods: *ROIs analysis*). *Figure 2F* shows that following these corrections the superficial bias in the GE-EPI acquired BOLD was significantly reduced. That is, the magnitude and variance of GE-EPI BOLD signals from voxels closer to the pial surface were reduced, as indicated by a significant interaction between GE-EPI acquired BOLD signal from different cortical depths (deeper, middle, superficial) before vs. after correction (F(2,28)=58.556, p<0.0001). That is, the superficial bias corrections resulted in decreased BOLD signal mainly in middle and superficial layers as indicated by post-hoc comparisons (middle: t = 7.992, p<0.0001; superficial: t = 11.241, p<0.0001).

Second, we scanned a subset of participants (N = 5) with a 3D GRASE sequence that is known to be sensitive to signals from small vessels and less affected by larger veins, resulting in higher spatial specificity of the measured BOLD signal (e.g. *De Martino et al., 2013*; *Kemper et al., 2015*). Consistent with previous studies (*De Martino et al., 2013*), the 3D GRASE data showed: (a) overall lower BOLD signal in V1 compared to the GE-EPI acquired BOLD data and (b) similar BOLD amplitude across V1 cortical depths. *Figure 2F* shows that the corrected GE-EPI BOLD signal in V1 follows a similar pattern across cortical depth as the 3D GRASE BOLD. In particular there were no significant differences in BOLD acquired with 3D GRASE vs. the corrected GE-EPI BOLD signal across cortical depths (i.e. no significant sequence × cortical depth interaction: F(2,6)=2.878, p=0.187), suggesting that our superficial bias corrections reduced substantially the contribution of vasculature-related signals in GE-EPI measurements. The small sample size does not allow further statistical analyses of the 3D GRASE data; yet, comparing the 3D GRASE and GE-EPI data provides a reproducibility test across MRI sequences. We observed similar fMRI adaptation patterns between sequences (*Figure 3—figure supplement 2*), suggesting that fMRI adaptation in superficial layers could not be simply attributed to vasculature-related confounds.

Taken together our results demonstrate fMRI adaptation across cortical depths of the visual cortex with stronger effects in superficial and middle than deeper layers. Comparing performance on the RSVP task during scanning across conditions showed that it is unlikely that these fMRI adaptation effects were due to differences in attention, as the RSVP task was similarly difficult across conditions. In particular, the mean performance across participants (adaptation condition: 62.7 ± 0.3%; non-adaptation condition 60.15 ± 0.4%, SEM) did not differ significantly between conditions (t(12) =0.312, p=0.76). Finally, we asked whether this layer-specific fMRI adaptation relates to perceptual bias in orientation discrimination due to adaptation (*Figure 3—figure supplement 1*), as measured by a tilt-aftereffect paradigm (e.g. *Gibson and Radner, 1937*). Correlating a perceptual adaptation index (i.e. difference in the perceived orientation of a test grating between adaptation and non-adaptation conditions) and fMRI adaptation (i.e. mean fMRI adaptation index across V1, V2, V3, and V4) showed a significant correlation for superficial (r = 0.656, p=0.039) but not middle

(r = 0.562, p=0.091), nor deeper (r = 0.567, p=0.087) layers. These results suggest that laminar-specific fMRI adaptation in visual cortex relates to perceptual bias in orientation discrimination due to adaptation.

## fMRI adaptation in intraparietal cortex

We next tested for adaptive processing in posterior parietal cortex regions (IPS1 IPS2; *Benson et al., 2014*; *Benson et al., 2012*; *Wang et al., 2015*) that have been shown to be involved in processing expectation due to stimulus familiarity (*de Lange et al., 2018*; *Garrido et al., 2009*; *Li et al., 2010*; *Summerfield and de Lange, 2014*). A repeated measures ANOVA (ROI [IPS1 IPS2], condition [adaptation non-adaptation], and cortical depth [deeper, middle, superficial]) showed a significant main effect of condition (i.e. decreased fMRI responses for adaptation compared to non-adaptation [F(1,14)=7.994, p=0.013]) and cortical depth (F(2,28)=25.824, p<0.0001). We did not observe any significant interactions between condition and cortical depth (F(2,28) =0.575, p=0.511), nor between ROI, condition, and cortical depth (F(2,28)=0.639, p=0.510), suggesting similar fMRI adaptation effects across cortical layers (*Figure 4*), rather than layer-specific fMRI adaptation, in posterior parietal cortex. Spatial regression analysis showed similar pattern of results (main effect of condition: F(1,14)=6.149, p<0.05, cortical layer: F(2,28)=22.359, p<0.0001), suggesting that our results were unlikely to be due to vasculature-related confounds.

## Functional connectivity

UHF fMRI allows us to interrogate the finer functional connectivity across areas based on known anatomical models of connectivity across cortical layers. Recent work (for review: *Lawrence et al., 2019a*) has proposed that anatomical connections between superficial V1 layers and middle layers of higher areas relate to feedforward processing, while anatomical connections between deeper V1 layers and deeper layers of higher areas relate to feedback processing (*Figure 1A*). We tested functional connectivity in these circuits to discern feedforward vs. feedback processing for orientation-specific adaptation. We did not test connectivity between superficial V1 layers and deeper layers of higher areas, as these connections are known to relate to both feedback and feedforward processing (*Maunsell and van Essen, 1983*; *Rockland and Virga, 1989*). Despite the fact that the UHF imaging resolution does not support one-to-one mapping between MRI-defined cortical depths and cyto-architectonically defined layers, *Figure 1A* provides a framework of feedback vs. feedforward connections across superficial, middle, and deeper cortical depths, as proposed and tested by

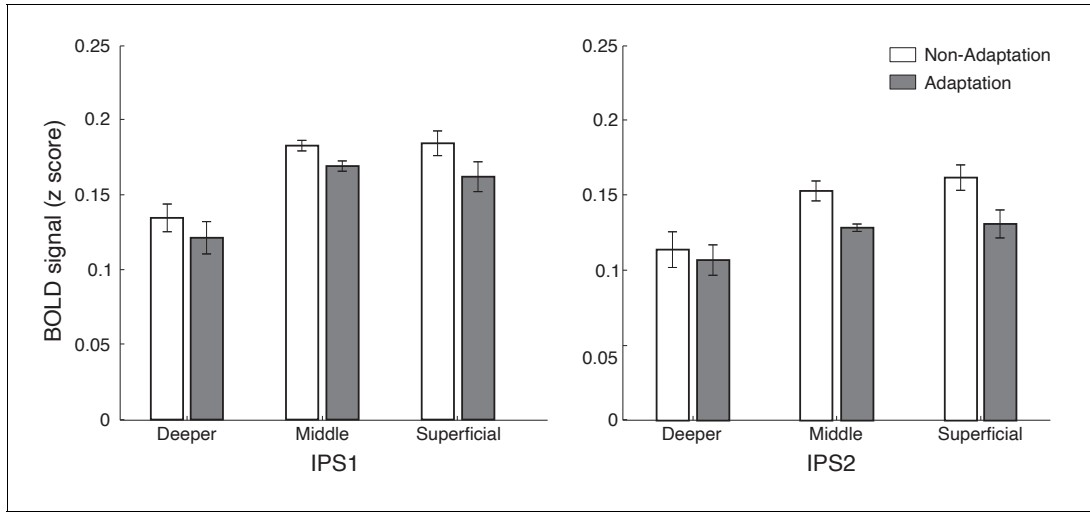

**Figure 4.** Laminar BOLD for IPS1 and IPS2. Mean BOLD in IPS1 and IPS2 across cortical layers. Bar-plots show z-scored BOLD signal for adaptation (grey) and non-adaptation (white) across cortical layers of IPS1 and IPS2. Error bars indicate within-subject confidence intervals (*Cousineau, 2005*) (N = 15).

The online version of this article includes the following source data for figure 4:

**Source data 1.** Tables for mean BOLD responses to adaptation and non-adaptation across cortical layers of IPS1 and IPS2.

previous UHF imaging studies (e.g. *Huber et al., 2017*; *Kok et al., 2016*; *Moerel et al., 2020*; *Sharoh et al., 2019*).

Using this framework, we computed functional connectivity within visual cortex and between visual and posterior parietal cortex. We used independent component analysis (ICA)-based denoising and finite impulse response (FIR) functions to denoise and deconvolve the fMRI time course data per cortical depth, controlling for noise and potential task-timing confounds. We then conducted Pearson correlations between the fMRI eigenvariate time courses across cortical depths. Our results (*Figure 5*) showed stronger feedforward connectivity for adaptation within visual cortex (i.e. V1 superficial layers and middle layers of higher visual areas), stronger short-range feedback connectivity between V2 and V1 deeper layers, and stronger long-range feedback occipito-parietal connectivity (i.e. V1 deeper layers and IPS1). In particular, a repeated measures ANOVA showed a significant three-way interaction ($F_{(4,52)}=3.574$, $p=0.027$) between connections (V1–V2, V1–V3, V1–V4, V1–IPS1, V1–IPS2), pathways (feedforward feedback), and condition (adaptation non-adaptation). Further, we tested whether the difference in connectivity between adaptation and non-adaptation (i.e. difference in Fisher z-transformed values) differed between pathways (feedforward feedback) and across connections (V1–V2, V1–V3, V1–V4, V1–IPS1, V1–IPS2). A repeated measures ANOVA showed a significant pathway (feedforward feedback) × connection (V1–V2, V1–V3, V1–V4, V1–IPS1, V1–IPS2) interaction ($F_{(4,52)}=3.945$, $p=0.022$).

For functional connectivity within visual cortex, post-hoc comparisons showed significantly higher connectivity between V1 superficial and middle layers of higher visual areas for adaptation compared to non-adaptation (V1–V2: $t(14)=2.324$, $p=0.036$; V1–V3: $t(14)=2.778$, $p=0.016$; V1–V4: $t(13)=2.778$, $p=0.0157$), suggesting enhanced feedforward processing for adaptation within visual cortex. In contrast, no significant differences between conditions were observed in functional connectivity between deeper layers in V1 and higher visual areas (V3: $t(14)=0.703$, $p=0.494$; V4: $t(13)=0.813$, $p=0.431$) with the exception of V1–V2 connectivity ($t(14)=2.223$, $p=0.043$), suggesting enhanced short-range feedback connectivity between V2 and V1 deeper layers for adaptation.

For occipito-parietal connectivity, we tested differences in connectivity between V1 layers and IPS subregions (IPS1 IPS2) as there were no significant differences in fMRI adaptation across IPS layers ($F_{(2,28)}=0.575$, $p=0.511$). Our results showed significantly higher functional connectivity between V1 deeper layers and IPS1 for adaptation compared to the non-adaptation ($t(14)=3.014$, $p=0.009$). This result is consistent with fMRI adaptation in V1 deeper layers ($t(14)=3.438$, $p=0.004$) and suggests enhanced feedback processing for visual adaptation. In contrast, no significant differences between conditions were observed for functional connectivity between: (a) V1 superficial layers and IPS1 (i.e.

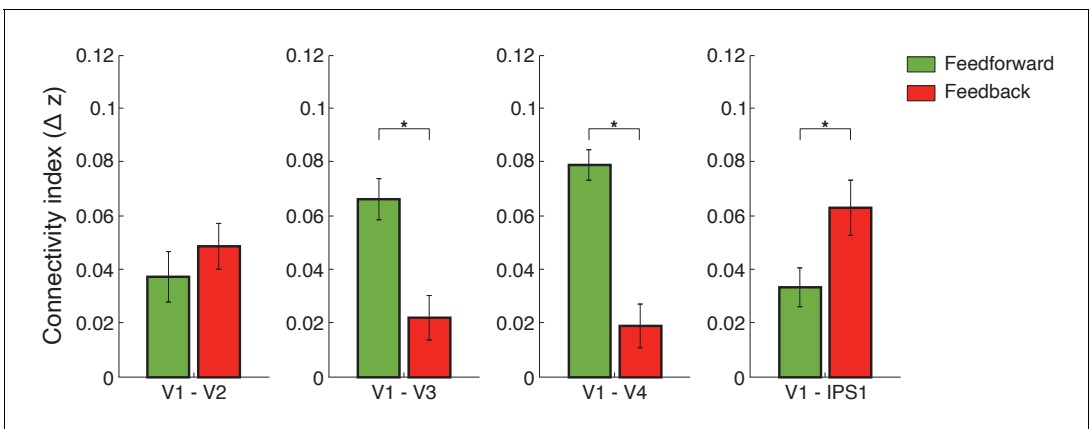

**Figure 5.** Functional connectivity. Bar-plots show difference (adaptation minus non-adaptation) in Fisher z-transformed r values for connectivity between V1 and V2, V1 and V3, V1 and V4, and V1 and IPS1. Feedforward connections were tested between: (a) V1 superficial and V2, V3, V4 middle layers (b) V1 superficial and IPS1. Feedback connections were tested between: (a) V1 deeper and V2, V3, V4 deeper layers (b) V1 deeper layers and IPS1. Error bars indicate within-subject confidence intervals (*Cousineau, 2005*) (N = 15 for V1, V2, V3, and IPS1; N = 14 for V4). Stars indicate statistically significant comparisons for p<0.05.

The online version of this article includes the following source data for figure 5:

**Source data 1.** Tables for cortical depth dependent values of feedforward and feedback functional connectivity.

functional connectivity related to feedforward processing) (t(14)=1.242, p=0.235), (b) V1 and IPS2 (V1 deeper layers and IPS2: t(14)=1. 243, p=0.234; V1 superficial layers and IPS2: t(14)=0.629, p=0.539), and (c) extrastriate visual areas and IPS regions (all p>0.146), suggesting specificity of feedback connectivity for adaptation between V1 and IPS1. Further, a regression analysis showed a significant relationship (r = 0.715, p=0.003) between fMRI adaptation (i.e. difference in z-scored BOLD between adaptation and non-adaptation) in deeper layers of V1 and deeper layers of IPS1. These results suggest that fMRI adaptation in deeper layers of IPS1 predicts fMRI adaptation in deeper layers of V1, consistent with enhanced feedback occipito-parietal connectivity for adaptive processing (i.e. top-down influences from IPS1 to processing of visual information in deeper V1 layers).

Finally, to test whether our results were specific to functional connectivity within the visual cortex and between V1 and IPS, we tested connectivity with hMT+ as a control region that is known to be anatomically connected to V1 (*Felleman and Van Essen, 1991*) but its function relates to motion rather than orientation processing. We defined hMT+ per participant using a probabilistic atlas (*Rosenke et al., 2020*) and segmented three cortical layers (superficial, middle, and deeper) (see *Methods, MRI data analysis: Segmentation and cortical depth sampling*). Analysing BOLD responses across cortical depths did not show significant fMRI adaptation. A repeated measures ANOVA did not show a significant main effect of condition (F(1,14)=3.874, p=0.069), cortical depth (F(2,28)=2.898, p=0.101), nor a significant cortical depth × condition interaction (F(2,28)=2.392, p=0.138). We then conducted functional connectivity analyses across cortical depths of V1 and hMT+. That is, we tested for differences between conditions (adaptation vs. non-adaptation) in functional connectivity between: (a) V1 superficial layers and hMT+ middle layers (i.e. feedforward connectivity) and (b) V1 and hMT+ deeper layers (i.e. feedback connectivity). A two-way repeated measures ANOVA did not show any significant differences in these functional connectivity pathways between conditions (i.e. no significant pathway × condition interaction: F(1,14)=0.475, p=0.502). These results suggest that changes in functional connectivity due to adaptation are specific to connectivity within visual cortex (early and extrastriate areas) and between V1 and IPS.

## Discussion

Here, we exploit UHF laminar fMRI to interrogate adaptive processing across cortical depth at a finer scale than afforded by standard fMRI methods. Previous studies have focussed on fMRI adaptation as a tool for interrogating selectivity at the level of large-scale neural populations for a given stimulus dimension. This is typically measured by recording fMRI responses to a test stimulus following the repeated presentation of stimuli that have similar or different dimensions to the test stimulus (e.g. *Engel, 2005*; *Fang et al., 2007*; *Fang et al., 2005*). In contrast, our study interrogates the mechanisms underlying adaptive processing, as a signature of short-term sensory plasticity, by measuring fMRI responses during stimulus repetition rather than responses to a test stimulus following adaptation. Employing this paradigm in combination with UHF laminar fMRI allows us to interrogate fMRI responses during stimulus repetition at a sub-millimetre resolution to gain insights into the circuit processes (feedforward vs. feedback) underlying adaptive processing in the human brain.

Our results demonstrate that visual adaptation involves recurrent processing of orientation information in visual cortex, as indicated by orientation-specific fMRI adaptation (i.e. BOLD decreases due to stimulus repetition) across cortical depths with stronger effects in superficial and middle than deeper layers. Our findings are consistent with a recent study (*Ge et al., 2020*) showing stronger fMRI responses in superficial V1 layers for visual adaptation in the context of the Flash Grab aftereffect. Our study extends beyond this finding to provide further insights into the functional circuit underlying visual adaptation. In particular, we provide evidence for distinct functional connectivity mechanisms for adaptive processing: feedforward connectivity within the visual cortex, indicating inherited adaptation from early to higher visual areas (e.g. *Larsson et al., 2016*; *Solomon and Kohn, 2014*), short-range feedback connectivity from V2 to V1, and long-range feedback connectivity from posterior parietal cortex to V1, reflecting top-down influences (i.e. expectation of repeated stimuli) on visual processing. Further, we demonstrate that orientation-specific fMRI adaptation in superficial layers of visual areas relates to perceptual bias in orientation discrimination due to adaptation, suggesting a link between recurrent adaptive processing and behaviour.

We interpret our results within a framework of feedback vs. feedforward connectivity across cortical depths (*Figure 1A*), as proposed by previous UHF imaging studies (e.g. *Huber et al., 2017*; *Kok et al., 2016*; *Moerel et al., 2020*; *Sharoh et al., 2019*). In particular, sensory inputs are known to enter the cortex at the level of the middle layer (middle layer 4) and output information is fed forward through the superficial layer (superficial layer 2/3). In contrast, feedback information is thought to be exchanged mainly between deeper layers (deep layer 5/6) (*Larkum, 2013*; *Markov et al., 2014*), as well as superficial layers (*Rockland and Virga, 1989*).

Neurophysiological studies have shown that this micro-circuit is involved in a range of visual recognition (*Self et al., 2013*; *van Kerkoerle et al., 2014*) and attention (*Buffalo et al., 2011*) tasks. Recent laminar fMRI studies provide evidence for the involvement of this circuit in the context of sensory processing (*De Martino et al., 2015*) and visual attention (*Fracasso et al., 2016*; *Lawrence et al., 2019b*; *Scheeringa et al., 2016*). Our results show fMRI adaptation across layers in visual cortex with stronger effects in superficial and middle than deeper layers. It is likely that adaptation alters processing of orientation information in middle layers that is then forwarded to superficial layers, consistent with recurrent processing for sensory adaptation. These signals are then forwarded to higher visual areas, as indicated by increased functional connectivity between V1 and higher visual areas. These results are consistent with previous neurophysiological studies showing that sensory adaptation is a fast form of plasticity (*Gutnisky and Dragoi, 2008*; *Whitmire and Stanley, 2016*; *Xiang and Brown, 1998*) and brain imaging studies showing that adapted BOLD responses in higher visual areas are inherited from downstream processing in V1 (*Ashida et al., 2012*; *Larsson et al., 2016*).

It is possible that orientation-specific adaptation is implemented in visual cortex via recurrent processing of signals across V1 columns (*Self et al., 2013*). Horizontal connections across V1 columns are known to mediate iso-orientation inhibition (*Malach et al., 1993*) that is suppression of neurons that are selective for the same orientation across columns. Iso-orientation inhibition is shown to be more pronounced in superficial layers and support orientation tuning (*Rockland and Pandya, 1979*). In particular, previous work has shown that horizontal connections between V1 columns primarily terminate in middle and superficial layers (*Rockland and Pandya, 1979*) and pyramidal cells in superficial layers make extensive arborisations within the same layer (*Douglas and Martin, 2007*). Consistent with this interpretation, previous neurophysiological studies have shown stronger decrease in neural population responses due to stimulus repetition in superficial layers of V1, while delayed adaptation effects in middle and deeper levels (*Westerberg et al., 2019*).

An alternative explanation is that BOLD effects in superficial layers reflect feedback processing (e.g. *Gau et al., 2020*; *Muckli et al., 2015*). Previous work has shown that synaptic input to superficial layers may result due to increase in feedback signals carried by neurons that have dendrites projecting to the superficial layers and their cell bodies in deeper layers (*Larkum, 2013*). Our results showing increased functional connectivity between V1 and V2 deeper layers suggest that short-range feedback from V2 contributes to orientation processing in V1, consistent with recurrent processing within visual cortex. Further, our results showing fMRI adaptation in deeper layers in V1 and increased functional connectivity between IPS and deeper V1 layers suggest that long-range feedback from the posterior parietal cortex contributes to adaptive processing in V1, consistent with the role of parietal cortex in expectation and prediction due to stimulus repetition. Recent fMRI studies focussing on higher visual areas have investigated the role of expectation in repetition suppression that – similar to sensory adaptation for simple stimulus features in early visual areas – is characterised by decreased BOLD responses to more complex stimuli (i.e. faces objects) in higher visual areas (*Grill-Spector et al., 2006*). In particular, *Summerfield et al., 2008* showed stronger repetition suppression in the lateral occipito-temporal cortex for identical stimulus pairs that were repeated frequently, providing evidence for a role of top-down influences (i.e. expectation) in repetition suppression and visual processing.

A framework for linking adaptive processing within visual cortex and feedback repetition suppression mechanisms due to expectation is proposed by the predictive coding theory (*Friston, 2005*; *Rao and Ballard, 1999*; *Shipp, 2016*). According to this framework, perception results from comparing feedback expectation and prediction signals in upstream regions with feedforward signals in sensory areas. When these signals match, the error (i.e. the difference between the prediction fed back and the incoming sensory input) is low; in contrast, when the expectation does not match with the sensory input, the prediction error is high resulting in increased neural responses for unexpected

compared to expected (i.e. repeated stimuli). *Bastos et al., 2012* have proposed a microcircuit model of predictive coding that combines excitatory and inhibitory properties of pyramidal neurons across cortical layers to account for prediction encoding, prediction errors, and modulation of incoming sensory inputs to minimise prediction error. Considering our findings in light of this model provides insights in understanding the circuit underlying adaptive processing in the human brain. It is likely that long-range top-down information (e.g. expectation signals from posterior parietal cortex) is fed back to the deeper layers of V1 and it is then compared with information available at the superficial layers (i.e. iso-orientation inhibition). A mismatch (i.e. prediction error) of signals (i.e. expectation of a repeated stimulus compared to the presentation of an unexpected stimulus) results in decreased fMRI responses for expected compared to unexpected stimuli. This is consistent with previous laminar imaging studies showing stronger fMRI responses in deeper or superficial V1 layers for perceptual completion tasks and suggesting top-down influences in visual processing (*Kok et al., 2016*; *Muckli et al., 2015*).

It is important to note that despite the advances afforded by UHF imaging, GE-EPI remains limited by vasculature-related signals contributing to BOLD at the cortical surface, resulting in loss of spatial specificity (*Kay et al., 2019*). To reduce this superficial bias, we removed voxels with low tSNR (*Olman et al., 2007*) and high t-statistic for stimulation contrast (*Kashyap et al., 2018*; *Polimeni et al., 2010*). Further, we applied a signal unmixing method (*Kok et al., 2016*; *Koster et al., 2018*) to control for draining vein effects from deep to middle and middle to superficial layers. We compared BOLD signals across conditions (adaptation vs. non-adaptation) and cortical depths after z-scoring the signals within each cortical depth to account for possible differences in signal strength across cortical layers (*Goense et al., 2012*; *Havlicek and Uludağ, 2020*). Following these corrections, we observed stronger fMRI adaptation (i.e. stronger BOLD response for non-adapted than adapted stimuli) in superficial layers, suggesting that our results are unlikely to be confounded by vasculature-related superficial bias. Further, using a 3D GRASE sequence that measures BOLD signals that are less affected by macro-vascular contribution showed similar results of layer-specific fMRI adaptation. Our findings on orientation-specific adaptation in superficial layers are consistent with previous laminar imaging studies showing BOLD effects in superficial layers in a range of tasks (*De Martino et al., 2015*; *Olman et al., 2012*). Recent advances in cerebral blood volume (CBV) imaging using vascular space occupancy (VASO) (e.g. *Beckett et al., 2020*; *Huber et al., 2019*) could be exploited in future studies to enhance the spatial specificity of laminar imaging in the human brain.

In sum, exploiting UHF imaging, we provide evidence that adaptive processing in the human brain engages a circuit that integrates recurrent processing within visual cortex with top-down influences (i.e. stimulus expectation) from posterior parietal cortex via feedback. This circuit of local recurrent and feedback influences is critical for rapid brain plasticity that supports efficient sensory processing by suppressing familiar and expected information to facilitate resource allocation to new incoming input. Combining laminar imaging with electrophysiological recordings has the potential to shed more light on the dynamics of this circuit, consistent with recent evidence (*Buffalo et al., 2011*; *Self et al., 2013*; *van Kerkoerle et al., 2014*) that gamma oscillations are linked to feedforward processing in input layers, while alpha/beta oscillations are related to feedback mechanisms in superficial and deeper cortical layers. Understanding these circuit dynamics is the next key challenge in deciphering the fast brain plasticity mechanisms that support adaptive processing in the human brain.

## Materials and methods

### Participants

Eighteen healthy volunteers (11 females and 7 males) participated in the study. Seventeen participants were scanned with a Gradient Echo-Echo Planar Imaging (GE-EPI) sequence (main experiment). Due to the lack of previous 7T fMRI adaptation studies, we determined sample size based on power calculations following a 3T fMRI study from our lab using the same paradigm (*Karlaftis et al., 2019*) that showed fMRI adaptation for effect size of Cohen's $f^2$ = 0.396 at 80% power. Data from two participants were excluded from further analysis due to excessive head movement (higher than 1.5 mm) and technical problems during acquisition, resulting in data from 15 participants for the

main experiment (mean age: 24.44 years and SD: 3.83 years). Five participants (four who participated in the main experiment and an additional participant) were scanned with a 3D GRASE EPI sequence. Twelve of the participants who took part in the fMRI experiment completed an additional psychophysical experiment. All participants had normal or corrected-to-normal vision, gave written informed consent, and received payment for their participation. The study was approved by the local Ethical Committee of the Faculty of Psychology and Neuroscience at Maastricht University and the University of Cambridge Ethics Committee (ethics number PRE2017.057).

## Stimuli

Stimuli comprised sinewave gratings (one cycle per degree) of varying orientations (*Figure 1B*). Stimuli were presented centrally within an annulus aperture (inner radius: 0.21°; outer radius: 6°). The outer edge of the aperture was smoothed using a sinusoidal function (standard deviation: 0.6°). Experiments were controlled using MATLAB and the Psychophysics toolbox 3.0 (*Brainard, 1997*; *Pelli, 1997*). For the main fMRI experiment, stimuli were presented using a projector and a mirror setup (1920 × 1080 pixels resolution, 60 Hz frame rate) at a viewing distance of 99 cm. The viewing distance was reduced to 70 cm for the control experiment, as a different coil was used, and adjusted so that angular stimulus size was the same for both scanning sessions.

## Experimental design

### fMRI session

Both the main and control fMRI experiments comprised a maximum of eight runs (13 participants completed eight runs for each experiment; two participants in the main experiment and one participant in the control completed six runs). Each run lasted 5 min 6 s, and started with 14 s fixation, followed by six stimulus blocks, three blocks per condition (adaptation non-adaptation) and ended with 14 s fixation. The order of the blocks was counterbalanced within and across runs. Each block comprised 16 stimuli followed by 2 s for response to the RSVP task. The orientation of the gratings was drawn randomly from uniform distributions, ranging from −85° to −5° and +5° to +85° in steps of 7.27°, excluding vertical (i.e. 0°). The same orientation was presented across adaptation blocks per participant. Sixteen different orientations were presented per block for the non-adaptation condition. Each stimulus was displayed for 1900 ms with a 100 ms inter-stimulus interval for both the adaptation and non-adaptation conditions to ensure similar stimulus presentation parameters (e.g. stimulus transients) between conditions. During scanning participants were asked to perform an RSVP task. A stream of letters was presented in rapid serial order (presentation frequency: 150 ms, asynchronous to the timings of grating presentation) within an annulus at the centre of the screen (0.5° of visual angle). Participants were asked to fixate at the annulus and report, by a key press at the end of each block, the number of targets (one to four per block). No feedback was provided to the participants.

In the same scanning session, anatomical data and fMRI data for retinotopic mapping were collected following standard procedures (e.g. *Engel et al., 1997*).

## MRI acquisition

Imaging data were acquired on a 7T Magnetom scanner (Siemens Medical System, Erlangen, Germany) at the Scannexus Imaging Centre, Maastricht, The Netherlands. Anatomical data were acquired using an MP2RAGE sequence (TR = 5 s, TE = 2.51 ms, FOV = 208×208 mm, 240 sagittal slices, 0.65 mm isotropic voxel resolution).

For the main experiment (N = 17), we used a 32-channel phased-array head coil (NOVA Medical, Wilmington, MA, USA) and a 2D Gradient Echo, Echo Planar Imaging (GE-EPI) sequence (TE = 25 ms, TR = 2 s, voxel size = 0.8 mm isotropic, FOV = 148×148 mm, number of slices = 56, partial Fourier = 6/8, GRAPPA factor = 3, Multi-Band factor = 2, bandwidth = 1168 Hz/Pixel, echo spacing = 1 ms, flip angle = 70°. The field of view covered occipito-temporal and posterior parietal areas; manual shimming was performed prior to the acquisition of the functional data.

For the control experiment (N = 5), participants were scanned with a 3D inner-volume gradient and spin echo (GRASE) sequence with variable flip angles (*Feinberg and Oshio, 1991*; *Kemper et al., 2016*). This sequence is largely based on a spin echo sequence for which the measured T2-weighted BOLD signal has higher spatial specificity and is less confounded by large

draining veins near the pial surface (e.g. *Duong et al., 2003*; *Goense et al., 2007*; *Kemper et al., 2015*; *Uludağ et al., 2009*). We used a custom-built surface-array coil (*Sengupta et al., 2016*) for enhanced SNR of high-resolution imaging of visual cortex (TR = 2 s, TE = 35.41 ms, FOV = 128×24 mm, number of slices = 12, echo-spacing = 1.01 ms, total readout train time = 363.6, voxel size = 0.8 mm isotropic, 90° nominal excitation flip angle, and variable refocussing flip angles ranging between 47° and 95°). The latter was used to exploit the slower decay of the stimulated echo pathway and hence to keep T2-decay-induced blurring in partition-encoding direction at a small, acceptable level, that is, comparable to the T2*-induced blurring in typical EPI acquisition protocols for functional imaging (*Kemper et al., 2016*).

## MRI data analysis
### Segmentation and cortical depth sampling
T1-weighted anatomical data was used for coregistration and 3D cortex reconstruction. Grey and white matter segmentation was performed on the MP2RAGE images using FreeSurfer (http://surfer.nmr.mgh.harvard.edu/) and manually improved for the ROIs (i.e. V1, V2, V3, V4, IPS, and hMT+) using ITK-Snap (http://www.itksnap.org/pmwiki/pmwiki.php, *Yushkevich et al., 2006*). The refined segmentation was used to obtain a measurement of cortical thickness. Following previous studies, we assigned voxels in three layers (deeper, middle, and superficial) using the equi-volume approach (*Kemper et al., 2018*; *Waehnert et al., 2014*) as implemented in BrainVoyager (Brain Innovation, Maastricht, The Netherlands). This approach has been shown to reduce misclassification of voxels to layers, in particular for ROIs presenting high curvature. Information from the cortical thickness map and gradient curvature was used to generate four grids at different cortical depths (ranging from 0, white matter, to 1, grey matter). Mapping of each voxel to a layer was obtained by computing the Euclidean distance of each grey matter voxel to the grids: the two closest grids represent the borders of the layer a voxel is assigned to (*Figure 2D*). Note that due to limitations in the UHF imaging resolution these MRI-defined layers indicate distance (i.e. cortical depth) from the grey matter/white matter and the grey matter/cerebrospinal fluid boundaries rather than one-to-one mapping to the cyto-architectonically defined layers of the human neocortex.

For the 3D GRASE control experiment, we used the LAYNII tools (*Huber et al., 2020*), as they provided better segmentation for images with a limited field of view.

## GE-EPI functional data analysis
The GE-EPI functional data were analysed using BrainVoyager (version 20.6, Brain Innovation, Maastricht, The Netherlands) and custom MATLAB (The MATHWORKS Inc, Natick, MA, USA) code. Preprocessing of the functional data involved three serial steps starting with correction of distortions due to non-zero off-resonance field; that is, at the beginning of each functional run, five volumes with inverted phase encoding direction were acquired and used to estimate a voxel displacement map that was subsequently applied to the functional data using COPE, BrainVoyager, Brain Innovation. The undistorted data underwent slice-timing correction, head motion correction (the single band image of each run was used as reference for the alignment), high-pass temporal filtering (using a GLM with Fourier basis set at two cycles), and removal of linear trends. Preprocessed functional data were coaligned to the anatomical data using a boundary-based registration approach, as implemented in BrainVoyager (Brain Innovation, Maastricht, The Netherlands). Results were manually inspected and further adjusted where needed. To validate the alignment of functional to anatomical data, we calculated the mean EPI image of each functional run for each ROI and estimated the spatial correlation between these images (e.g. *Marquardt et al., 2018*). We performed manual adjustment of the alignment if the spatial correlation was below 0.85. We excluded a small number of runs (N = 3 and N = 1 for two participants respectively), as their alignment could not be improved manually.

## 3D GRASE functional data analysis
Functional images were analysed using BrainVoyager (version 21.0, Brain Innovation, Maastricht, The Netherlands), custom MATLAB (The MATHWORKS Inc, Natick, MA, USA) code, and advanced normalisation tools (*Avants et al., 2011*) for registration of images. The first volume of each run was removed to allow for the magnetisation to reach a steady state. Head motion correction was

performed using as reference the first image (10 volumes with TR = 6 s) acquired at the beginning of the functional runs. The higher contrast of this image facilitated the coregistration of the anatomical and functional images. After motion correction, temporal high-pass filtering was applied, using a GLM with Fourier basis set at three cycles per run. Preprocessed images were converted into Nifti files and an initial manual registration was performed between the first image and the anatomical image using the manual registration tool provided in ITK-Snap (http://www.itksnap.org/pmwiki/pmwiki.php, *Yushkevich et al., 2006*). The resulting transformation matrix was applied to coregister the anatomical image to the functional space and fine-tuned adjustments were provided by means of antsRegistration tools.

## ROIs analysis

We used the data from the retinotopic mapping scan to identify ROIs. For each participant, we defined areas V1–V4 based on standard phase-encoding methods. Participants viewed rotating wedges that created travelling waves of neural activity (e.g. *Engel et al., 1997*). Due to limited coverage during acquisition, it was not possible to map area V4 in 1 of the 15 participants. Due to limited scanning time, it was not possible to perform an additional localiser scan to functionally identify posterior parietal cortex regions (comprising IPS1 and IPS2). We identified these regions based on a probabilistic atlas (https://hub.docker.com/r/nben/occipital_atlas/; *Benson et al., 2014*; *Benson et al., 2012*; *Wang et al., 2015*). This atlas defines IPS regions based on functional – rather than anatomical only – criteria based on previous work showing that these regions are involved in saccadic eye movements, spatial attention, and memory (e.g. *Schluppeck et al., 2005*; *Sereno et al., 2001*; *Silver and Kastner, 2009*; *Wang et al., 2015*). Finally, we defined hMT+ as a control ROI based on a functional atlas of visual cortex (*Rosenke et al., 2020*). For both the IPS regions and hMT+, we used the individual participant-based segmentation obtained with FreeSurfer and an anatomical probabilistic template to estimate the best location for the ROI (i.e. IPS). For each participant, we then visually inspected the ROI mapping to ensure good alignment and consistency across participants.

For each ROI and individual participant, we modelled BOLD signals using a block design GLM with two regressors, one per stimulus condition (adaptation = non-adaptation). We included estimated head motion parameters as nuisance regressors. The resulting t-statistical map was thresholded (t = 1.64, p=0.10) to select voxels within each ROI that responded more strongly to the stimulus conditions compared to fixation baseline (*Figure 2B and C*).

Voxel selection within each ROI was further refined by excluding voxels that were confounded by vasculature effects that are known to contribute to a superficial bias in the measured BOLD signal; that is, increased BOLD with increasing distance from white matter (see *Results: Control analyses*). In particular, it has been shown that the BOLD signal measured using GE-EPI, T2* weighted sequences is confounded by vasculature-related signals (*Uğurbil et al., 2003*; *Uludağ et al., 2009*; *Yacoub et al., 2005*) due to veins penetrating the grey matter and running through its thickness, as well as large pial veins situated along the surface of the grey matter (*Duvernoy et al., 1981*). This results in increased sensitivity (i.e. strong BOLD effect) but decreased spatial specificity of the measured signal.

Here, we took the following approach to reduce superficial bias due to vasculature contributions. First, following previous work (*Olman et al., 2007*), we computed tSNR for each voxel in each ROI (V1, V2, V3, V4, IPS1, and IPS2). We used this signal to identify voxels near large veins that are expected to have large variance and low intensity signal (mean tSNR across V1 smaller than 12.02 ± 2.02) due to the local concentration of deoxygenated haemoglobin resulting in a short T2* decay time. Second, it has been shown that high t-values on an fMRI statistical map are likely to arise from large pial veins (*Kashyap et al., 2018*; *Polimeni et al., 2010*). Therefore, voxels with t-score values above the 90th percentile (mean t-score across V1 larger than t = 15.47 ± 4.16) of the t-score distribution obtained by the GLM described above were removed from further analysis.

Further to account for possible differences in signal strength across cortical layers due to thermal and physiological noise, as well as signal gain (*Goense et al., 2012*; *Havlicek and Uludağ, 2020*), we (a) matched the number of voxels across cortical depths (i.e. to the layer with the lowest number of voxels) per participant and ROI, and (b) z-scored the time courses within cortical layer per ROI, controlling for differences in signal levels across cortical depths while preserving signal differences across conditions (after correction of vascular contributions, e.g. *Lawrence et al., 2019b*).

Normalised fMRI responses for each condition (adaptation non-adaptation) were averaged across the stimulus presentation (excluding participant responses; 32–34 s after stimulus onset), blocks, and runs for each condition. For visual cortex ROIs, we focussed on the time window that captured the peak of the haemodynamic response to visual stimulus presentation (4–18 s after stimulus onset). To test for layer-specific differences between conditions (adaptation vs. non-adaptation) across cortical depths and brain regions, we conducted repeated measures ANOVAs with condition (adaptation non-adaptation), cortical depth (deeper, middle, superficial layers), and ROIs as factors. Post-hoc comparisons (Bonferroni corrected for multiple comparisons) were used to further test differences in fMRI adaptation following significant ANOVA effects.

## Functional connectivity analysis

We followed standard analyses methods to compute functional connectivity across ROIs and cortical depths. We preprocessed the functional and anatomical data in SPM12.3 (v6906; http://www.fil.ion.ucl.ac.uk/spm/software/spm12/). We first performed brain extraction and normalisation to MNI space on the anatomical images (non-linear). The functional images were then corrected for distortions, slice-scan timing (i.e. to remove time shifts in slice acquisition), head motion (i.e. aligned each run to its single band reference image), coregistered all EPI runs to the first run (rigid body), coregistered the first EPI run to the anatomical image (rigid body), and normalised to MNI space (applying the deformation field of the anatomical images). Data were only resliced after MNI normalisation to minimise the number of interpolation steps.

Next, we used an ICA-based denoising procedure (*Griffanti et al., 2014*). We applied spatial smoothing (2 mm) and linear detrending, followed by spatial group ICA. The latter was performed using the Group ICA fMRI Toolbox (GIFT v3.0b) (http://mialab.mrn.org/software/gift/). Principal component analysis was applied for dimensionality reduction, first at the subject level, then at the group level. A fixed number (N = 35) of independent components was selected for the ICA estimation. The ICA estimation (Infomax) was run 20 times and the component stability was estimated using ICASSO (*Himberg et al., 2004*). Group information guided ICA back-reconstruction was used to reconstruct subject-specific components from the group ICA components (*Du et al., 2016*). The results were visually inspected to identify noise components according to published procedures (*Griffanti et al., 2017*). We labelled 12 of the 35 components as noise that captured signal from veins, arteries, cerebrospinal fluid pulsation, susceptibility, and multi-band artefacts.

To clean the fMRI signals from signals related to motion and the noise components, we followed the soft cleanup approach (*Griffanti et al., 2014*) on the BrainVoyager unsmoothed data in native space (see *GE-EPI functional data analysis*). That is, we first regressed out the motion parameters (translation, rotation, and their squares and derivatives; *Friston et al., 1996*) from each voxel and ICA component time course. Second, we estimated the contribution of every ICA component to each voxel's time course (multiple regression). Finally, we subtracted the unique contribution of the noise components from each voxel's time course to avoid removing any shared signal between neuronal and noise components.

Further, following recent work (*Cole et al., 2019*), we deconvolved the denoised time courses using FIR functions. In particular, we fitted 23 regressors per condition that covered the duration of each task block, including the response period and fixation block, to capture the whole haemodynamic response. This method allows us to accurately model and remove the cross-block mean response for each condition (adaptation and non-adaptation) to account for potential task-timing confounds that have been shown inflate the strength of the computed task-based functional connectivity. Within the GLM, the data were high-pass filtered at 0.01 Hz and treated for serial autocorrelations using the FAST autoregressive model (*Corbin et al., 2018*; *Olszowy et al., 2019*). For each ROI and cortical depth, we then computed the first eigenvariate across all voxels within the region to derive a single representative time course per cortical depth and ROI for connectivity analysis. We computed functional connectivity as the Pearson correlation between the eigenvariate time courses across ROIs and cortical depths. Finally, we performed repeated measures ANOVAs with connection (V1–V2, V1–V3, V1–V4, V1–IPS1, V1–IPS2), pathway (feedforward feedback), and condition (adaptation non-adaptation) as factors to test for differences in the functional connectivity values (after Fisher z-transform) between conditions. Post-hoc comparisons (Bonferroni corrected for multiple comparisons) were used to further test differences in functional connectivity following significant ANOVA effects.

## Acknowledgements

We would like to thank Christopher Wiggins and Esther Steijvers (Scannexus) for technical support, and Peter Kok (University College London), Denis Schluppeck (University of Nottingham), Federico De Martino (University of Maastricht), Laurentius Huber (University of Maastricht), and Cheryl Olman (University of Minnesota) for the expert and insightful comments to the manuscript. We would also like to thank Adrian Ng, Valentyna Chernova, and Cher Zhou for help with the analysis. This work was supported by grants to ZK from the Biotechnology and Biological Sciences Research Council (H012508 and BB/P021255/1) and was funded in part by the Wellcome Trust (205067/Z/16/Z). For the purpose of Open Access, the author has applied a CC BY public copyright licence to any Author Accepted Manuscript version arising from this submission.

## Additional information

### Funding

| Funder | Grant reference number | Author |
| --- | --- | --- |
| Biotechnology and Biological Sciences Research Council | H012508 | Zoe Kourtzi |
| Biotechnology and Biological Sciences Research Council | BB/P021255/1 | Zoe Kourtzi |
| Wellcome Trust | 205067/Z/16/Z | Zoe Kourtzi |

The funders had no role in study design, data collection and interpretation, or the decision to submit the work for publication.

### Author contributions

Elisa Zamboni, Conceptualization, Data curation, Software, Formal analysis, Validation, Investigation, Visualization, Methodology, Writing - original draft, Project administration, Writing - review and editing; Valentin G Kemper, Resources, Software, Investigation, Methodology, Writing - review and editing; Nuno Reis Goncalves, Resources, Software, Investigation, Methodology; Ke Jia, Resources, Software, Methodology; Vasilis M Karlaftis, Resources, Data curation, Software, Formal analysis, Methodology, Writing - review and editing; Samuel J Bell, Resources, Data curation, Software, Formal analysis, Writing - review and editing; Joseph Giorgio, Reuben Rideaux, Software, Writing - review and editing; Rainer Goebel, Conceptualization, Resources, Methodology, Writing - review and editing; Zoe Kourtzi, Conceptualization, Resources, Supervision, Funding acquisition, Investigation, Methodology, Writing - original draft, Project administration, Writing - review and editing

### Author ORCIDs

Elisa Zamboni https://orcid.org/0000-0001-9200-8031
Vasilis M Karlaftis http://orcid.org/0000-0003-1285-1593
Reuben Rideaux http://orcid.org/0000-0001-8416-005X
Zoe Kourtzi https://orcid.org/0000-0001-9441-7832

### Ethics

Human subjects: Participants gave written informed consent. The study was approved by the local Ethical Committee of the Faculty of Psychology and Neuroscience at Maastricht University and the University of Cambridge Ethics Committee (ethics number PRE2017.057).

### Decision letter and Author response

Decision letter https://doi.org/10.7554/eLife.57637.sa1
Author response https://doi.org/10.7554/eLife.57637.sa2

## Additional files

### Supplementary files
- Transparent reporting form

### Data availability

Source data have been provided for Figures 3, 4, and 5. Data can also be found on the Cambridge Data repository.

The following dataset was generated:

| Author(s) | Year | Dataset title | Dataset URL | Database and Identifier |
|---|---|---|---|---|
| Zamboni E, Kemper VG, Goncalves NR, Jia K, Karlaftis VM, Bell SJ, Giorgio J, Rideaux R, Goebel R, Kourtzi Z | 2020 | Fine-scale computations for adaptive processing in the human brain | https://doi.org/10.17863/CAM.60330 | Cambridge Data repository, 10.17863/CAM.60330 |

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
