## [Decision Letter]

**Acceptance summary:**

Sensory adaptation reflects short-term brain plasticity that optimizes the efficiency of information processing. The present study uses cutting-edge ultra-high field fMRI to examine cortical layer-specific neural basis for adaptation. Valuable new data with sub-millimeter resolution are provided to advance our understanding of mechanisms supporting adaptive processing in human visual cortex.

**Decision letter after peer review:**

Thank you for submitting your article "Fine-scale computations for adaptive processing in the human brain" for consideration by *eLife*. Your article has been reviewed by three peer reviewers, and the evaluation has been overseen by a Reviewing Editor and Chris Baker as the Senior Editor. The reviewers have opted to remain anonymous.

The reviewers have discussed the reviews with one another and the Reviewing Editor has drafted this decision to help you prepare a revised submission.

Summary:

This study investigates the neural mechanisms of adaptation in human visual cortex with ultra-high-field fMRI. The stimuli were gratings that either had the same orientation repeatedly presented (adaptation) or gratings of different orientation repeatedly presented (non-adaptation). Attention was maintained at fixation throughout with a rsvp task. A primary claim is that adaption is stronger in superficial depths in visual areas V1 through V4, but is not modulated with depth in IPS1 and IPS2. Functional connectivity analyses are used to assess the relative strength of feedforward and feedback connections between the regions studied during adaptation, indicating enhanced feedback connectivity from IPS to V1 and enhanced feedforward connectivity from V1 to V2, V3, and V4 during adaptation. The study combines cutting-edge imaging techniques with clear experimental design and careful analysis to make an important and valuable contribution to our understanding of mechanisms supporting adaptive processing in human visual cortex.

Essential revisions:

Assuming that a direct physiological measure (e.g., spikes) for the present study at this time is difficult, some additional psychophysical experiments would be needed to successfully address reviewer #3's first main concern. Adding some psychophysical experiments would also help to address comments from reviewer #2 better than just adding analyses and tempering claims.

Reviewer #1:

1) The study overall did not add much new evidence that may advance our understandings of feedforward and feedback processes in the visual adaptation, and the reported results were to some extent predictable from previous studies (see review, Lawrence et al., 2019; Self et al., 2019).

2) The authors claimed higher functional connectivity between V1 deeper layers and IPS, however, the reported analysis showed that the difference in adaptation and non-adaptation conditions between V1 deeper layers and IPS1 just reached the significance (p =.049), while that in V1 deeper layers and IPS2 was not significant (p =.281). I feel that these results were not strong enough for the authors make the solid conclusion on the significant difference between adaptive conditions. Also, I noticed that the IPS region was defined using anatomical templates, did the authors included functional localization scan for the IPS regions? In addition, the above analyses were between V1 deeper layers and the overall IPS1 and IPS2 respectively. According to the model (see Figure 1 in the manuscript), the feedback connection was between V1 deeper layers and IPS deeper layer, so the functional connectivity analyses should be conducted between the deeper layers of V1 and deeper layers of IPS1/IPS2. Given that the manuscript has reported the significant different neural responses in different layers of IPS1 and IPS2, so I suggest that the authors do additional analyses on the functional connectivity between V1 deeper layer and IPS1/IPS2 deeper layers, and the results would provide more specific and stronger evidence on the feedback connections in visual adaption.

Reviewer #2:

My concerns fall into two broad categories: 1) that the statistical tests employed in this work don't sufficiently support the authors' claims, and 2) that, even when properly analyzed, the data presented here are insufficient evidence for the circuit-level conclusions presented throughout the manuscript. The authors could improve the evidence and present interpretations that are more closely tied to the data.

Conceptual

1) The language used does not adequately clarify the differences (and potential lack of alignment) between cortical depths and cortical layers. For example, the claim that "UHF imaging affords the sub-millimetre resolution necessary to examine fMRI signals across cortical layers" is not strictly true. Even at the small voxel size used here, cortical curvature, variable thickness, and partial volume effects all make it extremely challenging to map cortical depth to physiologically-distinct cortical layers. This distinction is critical given the motivation of this work as dissecting feedforward and feedback projections.

Suggestion: avoid the term "layer" and instead use "depth", and clarify in the manuscript that inferences about cortical layers, and thus, alignment to existing anatomical models of feedforward and feedback projections is limited.

2) It's not immediately clear how one should combine the functional connectivity and adaptation results into a single framework for thinking about mechanisms of adaptation. Is it possible, for example, that other regions (besides IPS1, IPS2, V2, V3, and V4) feedback to V1 and contribute to suppression? Is the strength of the IPS feedback during the adaptation condition predictive of the amount of adaptive suppression? Conducting these additional analyses would strengthen the authors' claim that top-down feedbacks from the IPS contribute to the adaptive processing reported in visual cortex.

Suggestion: (i) Compare the degree of feedforward and feedback connectivity between conditions for a control region, e.g., hMT+, to demonstrate that IPS is uniquely (or especially) involved in these computations. (ii) Demonstrate that the amount of feedback from IPS correlates with the amount of adaptation in deeper cortical depths in V1 across individual subjects, and (iii) Elucidate it this feedback from IPS that is correlated with adaptation is specific to V1 or occurs to other visual areas that show adaptation (V2, V3, V4).

Data Analyses and Validation of Results

1) The reporting of ANOVA results of differences in adaptation across ROIs and depths throughout the paper is confusing. It is unclear whether a) separate ANOVAs were run to test each effect (main or interaction), or b) that the results of the ANOVAs are misreported. For example, consider the ANOVA reported which tests the effects of ROI, depth, and condition on z-scored BOLD responses. The degrees of freedom (3, 39), indicates that there are four levels to the factor of interest (four ROIs), and 40 total measurements. Where does 40 come from? If there are 15 subjects and each contributes 12 data points (4 ROIs x 3 depths), I would expect that the variance being analyzed is that of 15 x 12 = 180 data points. Then, the next result presented is the main effect of condition, which has a reported (1, 13) degrees of freedom, suggesting that a different model was fit.

Suggestion: Run a single ANOVA and report main effects and interaction terms from that analysis.

2) The authors claim that adaptation is stronger in superficial V1 than in other cortical depths in the Introduction but write "visual cortex" instead of "V1" or "primary visual cortex" elsewhere. It is unclear when the authors are claiming that the result applies to V1 and when it applies to V1 through V4 in aggregate. This is problematic for three reasons. First, the claim being made should be clear, and the Introduction and Discussion should reflect the scope (V1 or V1-V4) intended. Second, if the authors claim is that suppression is stronger in superficial V1, a post-hoc test with appropriate multiple comparisons correction is needed. The tests are insufficient in that they don't compare superficial to middle depths directly, and in that they aggregate across all four ROIs instead of testing V1 separately. Third, if the authors claim is that suppression is stronger in superficial V1 through V4, then the direct comparison of superficial to middle depths is still lacking. Furthermore, the framing of the rest of the paper which compares V1 connectivity to IPS and V2-V4 and discusses V1 in the Introduction and Discussion needs to be justified if V1 is no different from V2, V3, and V4.

Suggestion: Make the claim being made clearer, then compute the appropriate statistical tests to support that claim. If needed, revise the Introduction and Discussion to explain special attention paid to each ROI.

3) In relation to point (2) above, they claim that adaptation is layer specific rests on the results of post-hoc tests. However, it is unclear (i) which ROIs are tested, and (ii) why they are comparing superficial and deep as well as middle and deep but not superficial and middle. Further the Materials and methods indicate that pairwise t-tests were used, if so multiple comparisons correction is needed to validate the result.

Suggestion: The authors should conduct all tests and clarify the reporting of the results. If multiple-comparisons correction is not currently being used, the authors should employ either Tukey's Honest Significant Difference, Bonferroni correction, or an alternative correction. If the tests are already corrected, that fact should be reflected in the Materials and methods section.

4) The inclusion of GRASE results strengthens the work substantially. However, the claim that the same adaptation patterns are observed are not supported numerically. In fact, the results presented in Figure 3—figure supplement 2 suggest that adaptation is not stronger in superficial than middle or deep depths, and that the correlation between adaptation indices across scan sequences (panel B) are moderate; at most, the adaptation indices from one scan sequence predicts 22% of the variance in the adaptation indices from the other scan sequence.

Suggestion: Report statistics for the 3D GRASE results.

Reviewer #3:

The motivation and framing is to characterize "circuit properties of adaptation" but I have two issues with this. First, is the inference that neural adaptation is occurring. Yes, the signal is smaller when stimuli are repeated vs. when they are not, and this is *consistent* with adaptation. But, the origin(s) of fMRI repetition effects is controversial. Without a secondary measure – e.g., psychophysical evidence of adaptation (e.g., reduced sensitivity, tilt aftereffect, etc) in the adapted vs. non-adapted conditions or a direct physiological measure (e.g., spikes) – all we can say is that the fMRI signal is reduced during repetition. What I think this paper does a great job of doing is the set up for an experiment that specifically examines adaptation – for example, is there a specific layer-response that best predicts psychophysical differences in adaptation?

The second issue is the inferences that are made between depth and feedback/feedforward processing. Take, for example, a measured difference in superficial layers. I don't understand how it is possible to know whether a change in the BOLD signal in superficial layers is due to neurons in these layers being affected by within-area circuits (e.g., from known connections between middle-layers to superficial layers) or due to feedback-mediated effects (as feedback affects both superficial and deep layers). In light of this, it's difficult to parse the first paragraph of the Discussion. "First, visual adaptation is implemented by recurrent processing of signals in visual cortex, as indicated by fMRI adaptation (i.e. BOLD decrease due to stimulus repetition) across layers with stronger effects in superficial than middle and deeper layers." What does "recurrent processing" mean here? If it means "feedback" shouldn't deeper layers have stronger effects? Does it mean with-area processing (middle/input -> superficial/output). Overall, I find the report of layer-specific responses interesting but I cannot infer the level of "circuit properties" the authors' wish to ascribe to the effects.

Along similar lines, as a way to refresh my memory of layers/connections in early visual cortex, I looked at this recent paper: "Anatomy and Physiology of Macaque Visual Cortical Areas V1, V2, and V5/MT: Bases for Biologically Realistic Models". It is difficult to map the relatively simple characterization presented in the current paper with the real complexity described in the Vanni et al. paper. As just one example, from Vanni et al., "FF connections from V1 to V5 arise from layers 4B (both blobs and interblobs) and 6 and target primarily L4 and less so L3 of V5." "FB projections from V5 to V1 terminate predominantly in layers 4B and 6 (Maunsell and Van Essen 1983b; Ungerleider and Desimone 1986b; Shipp et al. 1989), that is, the source layers of the V1-to-V5 FF projection." There is very little consistency between this characterization and what is depicted in Figure 1A (though, I understand these quotes are specifically about MT-V1 connections).

Reviewer #2:

1) "while feedback occipito-parietal connectivity" should be "while feedback was enhanced for occipito-parietal connectivity".

2) Introduction second sentence: This claim is lacking citations

3) Results final paragraph: why isn't IPS considered visual cortex? It's definition in the Wang et al. atlas is based on topographic representation of visual space.

Reviewer #3:

Introduction paragraph 1 and paragraph 2: plethora. Maybe don't use plethora twice.

Results: It seems like all ROIs should be in Figure 3A and remove Supplementary figure 3. All ROIs are in Figure 3B and all are included in the ANOVA. It would just be easier to refer to a figure that included all ROIs if ROIs are discussed in the ANOVA.

"Post-hoc comparisons showed significantly decreased fMRI responses for adaptation across cortical layers (deeper: t(14)=-3.244, p=0.006; middle: 126 t(14)=-3.920, p=0.002; superficial: t(14)=-4.134, p=0.001)." What are the t-tests comparing? Is that just in V1?

Error bars: Unfortunately, the error bars make it look as if there are no effects. Visually, looking at Figure 3A, I'd say the responses are identical across layers. And, I'd conclude the same about all ROIs in Figure 3B. In fact, as I write this, it's difficult to reconcile the p-values in the text and what is presented in the figures. I'm guessing the issue is the repeated-measures nature of the analysis in which case between-subject error bars are misleading. You might consider:

https://www.researchgate.net/profile/Denis_Cousineau/publication/49619408_Confidence_intervals_in_within-subject_designs_A_simpler_solution_to_Loftus_and_Masson's_method/links/54e4c9300cf22703d5bf6023.pdf

---

## [Author Response]

Essential revisions:Assuming that a direct physiological measure (e.g., spikes) for the present study at this time is difficult, some additional psychophysical experiments would be needed to successfully address reviewer #3's first main concern. Adding some psychophysical experiments would also help to address comments from reviewer #2 better than just adding analyses and tempering claims.

We thank the reviewers and the Editor for their constructive feedback. We agree that neurophysiology experiments are beyond the remit of this study that focuses on understanding the human brain computations that underlie adaptive processing. Following the reviewers’ suggestion, we have conducted a behavioural study employing a classic tilt after effect paradigm that has been used extensively to study orientation adaptation. Our results show that perceptual adaptation as measured by this paradigm correlates significantly with fMRI adaptation in superficial layers of visual areas, suggesting that the layer-specific fMRI adaptation we observed relates to adaptive behaviour.

Reviewer #1:1) The study overall did not add much new evidence that may advance our understandings of feedforward and feedback processes in the visual adaptation, and the reported results were to some extent predictable from previous studies (see review, Lawrence et al., 2019; Self et al., 2019).

We have revised the Introduction and Discussion sections to clarify the advances provided by our study in understanding the feedforward vs. feedback processes involved in visual adaptation using UHF laminar fMRI.

First, previous studies have used laminar fMRI to discern feedforward vs. feedback processes involved in a range of cognitive functions (e.g., attention, memory, visual motion, multisensory processing, tonotopic maps, visual imagery and illusions, to name just a few; De Martino et al., 2015; Fracasso, Petridou and Dumoulin, 2016; Lawrence, Norris and de Lange, 2019; Scheeringa et al., 2016). However, to the best of our knowledge, no previous study had investigated the contributions of these processes to sensory adaptation and plasticity due to stimulus repetition.

Second, previous studies focused on fMRI adaptation as a tool for interrogating selectivity at the level of large-scale neural populations for a given stimulus dimension. This is typically measured by recording fMRI responses to a test stimulus following the repeated presentation of stimuli that have similar or different dimensions to the test stimulus (e.g. Engel, 2005; Fang, Murray and He, 2007; Fang et al., 2005). In contrast, our study interrogates the mechanisms underlying adaptive processing, as a signature of short-term sensory plasticity, by measuring fMRI responses during stimulus repetition rather than responses to a test stimulus following adaptation. Employing this paradigm in combination with UHF laminar fMRI allows us to interrogate fMRI responses during stimulus repetition at a submillimetre resolution to gain insights into the circuit processes (feedforward vs. feedback) underlying adaptive processing in the human brain.

After receiving the reviews on our manuscript, a new study was published (Ge et al., 2020), using laminar fMRI to investigate the mechanisms underlying visual adaptation in the context of the Flash Grab aftereffect. The paradigm involves adaptation to wedged discs rotating clockwise and anti-clockwise with vertical bars presented on the disc’s wedge boundary at the time of rotation reversal. This study reports stronger fMRI responses in superficial V1 layers, as measured by the difference in mean BOLD response between clockwise and anti-clockwise adaptors conditions. At first glance, this result is similar to our finding showing laminar-specificity of fMRI adaptation in V1. Yet, our study extends beyond this finding to provide further insights into the functional circuit processes underlying visual adaptation. In comparison to Ge et al., 2020, our study advances our understanding in the following respects:

1) We interrogated laminar fMRI specificity for adaptation in regions beyond V1: a) across visual cortex (V1 and higher visual areas) involved in sensory processing, b) posterior parietal cortex known to be involved in expectation of familiar stimuli.

2) Combining functional connectivity analyses with laminar fMRI, we provide clearer insights in interpreting the laminar fMRI specificity by testing for connectivity that relates to feedforward (i.e. between superficial and middle layers across regions) vs. feedback (i.e. across deeper layers) processing. These analyses demonstrate the following novel findings related to the fine-scale circuit involved in adaptive processing: a) feedforward processing within visual cortex from V1 to higher visual areas b) recurrent processing as indicated by short-range feedback from V2 to V1, c) long-range feedback from IPS to V1.

3) We demonstrate that laminar specificity of fMRI adaptation in visual cortex relates to perceptual bias in orientation discrimination due to adaptation. Using a tilt aftereffect paradigm, we measured perceptual adaptation on the same individuals who participated in the fMRI study and demonstrated a significant correlation between fMRI adaptation in superficial layers of visual areas and perceptual bias, suggesting that processing in superficial visual cortex layers supports perceptual adaptation. These results strengthen the evidence we provide through a range of controls that the laminar fMRI specificity we report is unlikely to be due to vasculature-related confounds.

We now discuss these points and the Ge et al., 2020, study in the revised manuscript.

2) The authors claimed higher functional connectivity between V1 deeper layers and IPS, however, the reported analysis showed that the difference in adaptation and non-adaptation conditions between V1 deeper layers and IPS1 just reached the significance (p =.049), while that in V1 deeper layers and IPS2 was not significant (p =.281). I feel that these results were not strong enough for the authors make the solid conclusion on the significant difference between adaptive conditions.

We thank the reviewer for raising this point. We checked the localisation of the IPS ROIs and improved the coverage for 4 participants. Performing the connectivity analysis with these modified ROIs showed a clearer result of enhanced feedback connectivity between V1 deeper layers and IPS1 for adaptation compared to non-adaptation (p=0.009). There was no significant difference between conditions for connectivity between V1 deeper layers and IPS2, suggesting specificity of feedback from posterior IPS to V1.

Also, I noticed that the IPS region was defined using anatomical templates, did the authors included functional localization scan for the IPS regions?

To localise visual areas, we used a standard retinotopic mapping protocol using a checkerboard stimulus. Previous studies have reported that mapping regions in the posterior parietal cortex requires participants to engage in saccadic eye movements, spatial attention, or memory tasks (see Wang et al., 2015; Silver and Kastner, 2009; Schluppeck, Glimcher and Heeger, 2005; Sereno, Pitzalis and Martinez, 2001). Due to time constraints on the fMRI scans, we were not able to include a specific IPS functional localiser. Instead, we used the Atlas provided by Benson, based on the work from Wang et al., 2015, to identify subregions of the intraparietal sulcus. IPS ROIs in this atlas are based on a functional –rather than anatomical– definition following memory-guided saccade mapping. We have now clarified this in the revised manuscript.

In addition, the above analyses were between V1 deeper layers and the overall IPS1 and IPS2 respectively. According to the model (see Figure 1 in the manuscript), the feedback connection was between V1 deeper layers and IPS deeper layer, so the functional connectivity analyses should be conducted between the deeper layers of V1 and deeper layers of IPS1/IPS2. Given that the manuscript has reported the significant different neural responses in different layers of IPS1 and IPS2, so I suggest that the authors do additional analyses on the functional connectivity between V1 deeper layer and IPS1/IPS2 deeper layers, and the results would provide more specific and stronger evidence on the feedback connections in visual adaption.

We thank the reviewer for this suggestion. We averaged the signal across layers in IPS1 and IPS2, as we did not observe laminar-specificity in IPS (i.e. there was no significant differences in fMRI-adaptation across IPS layers). Following the reviewer’s suggestion, we tested for feedback connectivity between deeper V1 and IPS1 or IPS2 layers. Correlating fMRI adaptation index (i.e. difference in z-scored BOLD signal between adaptation and nonadaptation) between deeper layers of V1 and IPS1 showed a significant correlation (r=0.715, *p*=0.003), suggesting that feedback from deeper IPS1 layers contributes to adaptation in deeper V1 layers. The same analysis between deeper layers of IPS2 and V1 did not show a significant correlation (r=0.427, *p*=0.113), suggesting specificity of feedback from IPS1 to V1.

Reviewer #2:My concerns fall into two broad categories: 1) that the statistical tests employed in this work don't sufficiently support the authors' claims.

We have substantially revised our statistical tests, adding more information on the ANOVA models and clarifying the main and post-hoc tests performed, as well as the related corrections for multiple comparisons (Bonferroni correction). Further, we added a substantial section of control analyses. In addition, we have described the analyses across areas in the visual cortex, rather than simply V1, and have clarified both significant and non-significant tests.

2) Even when properly analyzed, the data presented here are insufficient evidence for the circuit-level conclusions presented throughout the manuscript. The authors could improve the evidence and present interpretations that are more closely tied to the data.

Following the reviewers’ suggestions, we have provided additional analyses and tests of functional connectivity, strengthening the evidence for the finer scale connectivity supporting adaptive processing. In addition, we have conducted a behavioural experiment, providing evidence that the fMRI laminar-specificity we observed (i.e. stronger fMRI adaptation in superficial V1 layers) relates to perceptual bias.

Conceptual1) The language used does not adequately clarify the differences (and potential lack of alignment) between cortical depths and cortical layers. For example, the claim that "UHF imaging affords the sub-millimetre resolution necessary to examine fMRI signals across cortical layers" is not strictly true. Even at the small voxel size used here, cortical curvature, variable thickness, and partial volume effects all make it extremely challenging to map cortical depth to physiologically-distinct cortical layers. This distinction is critical given the motivation of this work as dissecting feedforward and feedback projections.Suggestion: avoid the term "layer" and instead use "depth", and clarify in the manuscript that inferences about cortical layers, and thus, alignment to existing anatomical models of feedforward and feedback projections is limited.

We followed the terminology used in recent UHF imaging studies that often use different terms (e.g., cortical layers, cortical depth, mesoscopic scale, laminae, …) interchangeably (see, for example: https://layerfmri.com/2019/02/21/terminology/). However, we agree with the reviewer that UHF imaging does not support one-to-one mapping between MRI-defined cortical depths and cyto-architectonically defined cortical layers. As we describe in the manuscript, voxels within the grey matter/white matter and grey matter/CSF borders were assigned to three different subregions corresponding to superficial, middle and deeper layers. Following the reviewer’s suggestion, we have clarified this limitation of UHF imaging and have replaced “layers” with “cortical depths” where possible.

2) It's not immediately clear how one should combine the functional connectivity and adaptation results into a single framework for thinking about mechanisms of adaptation. Is it possible, for example, that other regions (besides IPS1, IPS2, V2, V3, and V4) feedback to V1 and contribute to suppression? Is the strength of the IPS feedback during the adaptation condition predictive of the amount of adaptive suppression? Conducting these additional analyses would strengthen the authors' claim that top-down feedbacks from the IPS contribute to the adaptive processing reported in visual cortex.Suggestion:i) Compare the degree of feedforward and feedback connectivity between conditions for a control region, e.g., hMT+, to demonstrate that IPS is uniquely (or especially) involved in these computations.

We thank the reviewer for this suggestion. Our choice of ROI was: a) motivated by our aim to test the fine scale computations that support adaptive processing in sensory areas and IPS that is known to be involved in expectation of familiar stimuli, b) limited by the coverage afforded when acquiring data at sub millimetre resolution. Following the reviewer’s suggestion, we tested hMT+ as a control ROI. Briefly, we used a probabilistic atlas (Rosenke et al., 2020) to obtain a functionally defined ROI for hMT+ for each participant. We manually adjusted the segmentation for this ROI and further assigned grey matter voxels within this ROI in three groups, generating cortical layers as described in more detail in the manuscript (see Materials and methods, MRI data analysis: Segmentation and cortical depth sampling). First, we extracted z-scored BOLD responses for each condition and cortical depth. A repeated measures ANOVA with cortical depth (deeper, middle, superficial) and condition (adaptation, non-adaptation) as factors showed no significant effect of cortical depth (F(2,28)=2.898, p=0.101), nor condition (F(1,14)=3.874, p=0.069), nor a significant interaction between these two factors (F(2,28)=2.392, p=0.138). Second, we performed laminar functional connectivity analysis as described in the revised manuscript (see Materials and methods, Functional Connectivity analysis) between V1 and hMT+. A repeated measures ANOVA with pathway (feedforward, feedback) and condition (adaptation, non-adaptation) as factors showed a main effect of condition (F(1,14)=5.828, p=0.03), but no significant effect of pathway nor an interaction between condition and pathway (F(1,14)=2.597, p=0.129; F(1,14)=0.475, p=0.502, respectively). These analyses suggest specificity of functional connectivity between V1 and IPS, i.e. fMRI adaptation in V1 deeper layers was due to feedback from IPS rather than other visual areas (e.g. hMT+).

ii) Demonstrate that the amount of feedback from IPS correlates with the amount of adaptation in deeper cortical depths in V1 across individual subjects.

We thank the reviewer for this suggestion. Our results showed stronger feedback connectivity for adaptation (i.e. connectivity across deeper layers) between: a) V1 and V2 (paired t-test comparing feedback connectivity between adaptation and non-adaptation: t(14)=2.223, *p*=0.043; no significant differences for V1-V3 connectivity: t(14)=0.703, *p*=0.494; nor V1V4 connectivity: t(13)=0.813, *p*=0.431), b) V1 and IPS (t(14)=3.014, *p*=0.009). Following the reviewer’s suggestion, we tested whether this feedback connectivity correlates with adaptation in deeper V1 layers. In particular, we correlated the fMRI adaptation index (i.e. difference in z-scored BOLD signal between adaptation and non-adaptation) between deeper layers of V1 and IPS1. We observed a significant correlation (r=0.715, *p*=0.003), suggesting that feedback from deeper IPS1 layers contributes to adaptation in deeper V1 layers. We have now included this analysis in the revised manuscript.

iii) Elucidate it this feedback from IPS that is correlated with adaptation is specific to V1 or occurs to other visual areas that show adaptation (V2, V3, V4).

We thank the reviewer for this suggestion and have now included additional statistics in the revised manuscript. In particular, feedback (i.e. correlation across deeper layers) from IPS1 was specific to V1. There was no significant correlation across deeper layers between a) IPS1 and V2, V3, or V4; that is, repeated measures ANOVA with pathway (feedforward, feedback), condition (adaptation, non-adaptation) showed no significant interactions between these factors; IPS1-V2: F(1,14)=0.241, *p*=0.631; IPS1-V3: F(1,14)=0.039, *p*=0.846; IPS1-V4: F(1,13)=2.387, *p*=0.146), b), IPS2 and V1, V2, V3, or V4; that is, repeated measures ANOVA with pathway (feedforward, feedback), condition (adaptation, non-adaptation) showed no significant interactions between these factors; IPS2V1: F(1,14)= 0.714, *p*=0.412; IPS2-V2: F(1,14)=0.386, *p*=0.545; IPS2-V3: F(1,14)=0.49, *p*=0.495; IPS2-V4: F(1,13)=0.264, *p*=0.616).

Data Analyses and Validation of Results1) The reporting of ANOVA results of differences in adaptation across ROIs and depths throughout the paper is confusing. It is unclear whether a) separate ANOVAs were run to test each effect (main or interaction), or b) that the results of the ANOVAs are misreported. For example, consider the ANOVA reported which tests the effects of ROI, depth, and condition on z-scored BOLD responses. The degrees of freedom (3, 39), indicates that there are four levels to the factor of interest (four ROIs), and 40 total measurements. Where does 40 come from? If there are 15 subjects and each contributes 12 data points (4 ROIs x 3 depths), I would expect that the variance being analyzed is that of 15 x 12 = 180 data points. Then, the next result presented is the main effect of condition, which has a reported (1, 13) degrees of freedom, suggesting that a different model was fit.Suggestion: Run a single ANOVA and report main effects and interaction terms from that analysis.

We apologise for the confusion and we have now revised the presentation of the results, as well as clarified the ANOVA models in the Materials and methods and Results sections. Please note that the definition of area V4 was not possible for one of the participants due to limited coverage. As a result, the ANOVA model had missing data (n=14 rather than n=15 for V4) and three main factors, ROI (V1, V2, V3, V4), cortical depth (deeper, middle, superficial), and condition (adaptation, nonadaptation).

That is degrees of freedom were calculated as follows: ROI with [(4-1),[(4-1)*(14-

1)]]=(3,39) degrees of freedom; cortical depth with [(3-1),[(3-1)*(14-1)]]=(2,26) degrees of freedom; condition with [(2-1),[(2-1)*(14-1)]]=(1,13) degrees of freedom. Accordingly, the interactions between factors were: ROI x cortical depth [(4-1)*(3-1), (4-1)*(3-1)*(141)]=(6,78); ROI x condition [(4-1)*(2-1), (4-1)*(2-1)*(14-1)]=(3,39); cortical depth x condition [(3-1)*(2-1), (3-1)*(2-1)*(14-1)]=(2,26); ROI x cortical depth x condition [(41)*(3-1)*(2-1), (4-1)*(3-1)*(2-1)*(14-1)]=(6,78).

2) The authors claim that adaptation is stronger in superficial V1 than in other cortical depths in the Introduction but write "visual cortex" instead of "V1" or "primary visual cortex" elsewhere. It is unclear when the authors are claiming that the result applies to V1 and when it applies to V1 through V4 in aggregate. This is problematic for three reasons. First, the claim being made should be clear, and the Introduction and Discussion should reflect the scope (V1 or V1-V4) intended. Second, if the authors claim is that suppression is stronger in superficial V1, a post-hoc test with appropriate multiple comparisons correction is needed. The tests are insufficient in that they don't compare superficial to middle depths directly, and in that they aggregate across all four ROIs instead of testing V1 separately. Third, if the authors claim is that suppression is stronger in superficial V1 through V4, then the direct comparison of superficial to middle depths is still lacking. Furthermore, the framing of the rest of the paper which compares V1 connectivity to IPS and V2-V4 and discusses V1 in the Introduction and Discussion needs to be justified if V1 is no different from V2, V3, and V4.Suggestion: Make the claim being made clearer, then compute the appropriate statistical tests to support that claim. If needed, revise the Introduction and Discussion to explain special attention paid to each ROI.

We thank the reviewer for these suggestions and have revised the manuscript accordingly.

First, we have clarified when we refer to V1 (primary visual cortex) vs. extrastriate visual areas (V2, V3, V4). We have revised the Introduction and Discussion to include all visual areas (not only V1) and IPS.

Second, we have added post-hoc comparisons (Bonferroni corrected for multiple comparisons) showing significantly stronger fMRI adaptation (i.e. fMRI responses for nonadaptation minus adaptation) across visual areas in superficial compared to deeper layers (V1: t(14)=-2.556, *p*=0.023; V3: t(14)=-2.580, *p*=0.022; V4: t(13)=-2.091, *p*=0.012) and middle compared to deeper layers (V1: t(14)=-2.429, *p*=0.029; V2: t(14)=-2.524, *p*=0.024; V3: t(14)=-3.528, *p*=0.003; V4: t(13)=-2.519, *p*=0.026). No significant differences were observed between fMRI adaptation in superficial and middle layers across visual areas (V1: t(14)=-2.093, *p*=0.055; V2: t(14)=0.331, *p*=0.746; V3: t(14)=0.942, *p*=0.362; V4: t(13)=2.096, *p*=0.056), nor between superficial and deeper layers of V2 (t(14)=-0.881, *p*=0.393).

Third, we have now added statistics showing no significant changes in functional connectivity between extrastriate and intraparietal areas and included additional discussion on the connectivity results.

3) In relation to point (2) above, they claim that adaptation is layer specific rests on the results of post-hoc tests. However, it is unclear (i) which ROIs are tested, and (ii) why they are comparing superficial and deep as well as middle and deep but not superficial and middle. Further the Materials and methods indicate that pairwise t-tests were used, if so multiple comparisons correction is needed to validate the result.Suggestion: The authors should conduct all tests and clarify the reporting of the results. If multiple-comparisons correction is not currently being used, the authors should employ either Tukey's Honest Significant Difference, Bonferroni correction, or an alternative correction. If the tests are already corrected, that fact should be reflected in the Materials and methods section.

We thank the reviewer for this suggestion. We had performed Bonferroni correction for multiple comparisons. We have now clarified this in the revised Materials and methods section. Further, we have clarified the factors compared for each test in the revised Results section. We have also added the non-significant results for comparisons between superficial and middle layers.

4) The inclusion of GRASE results strengthens the work substantially. However, the claim that the same adaptation patterns are observed are not supported numerically. In fact, the results presented in Figure 3—figure supplement 2 suggest that adaptation is not stronger in superficial than middle or deep depths, and that the correlation between adaptation indices across scan sequences (panel B) are moderate; at most, the adaptation indices from one scan sequence predicts 22% of the variance in the adaptation indices from the other scan sequence.Suggestion: Report statistics for the 3D GRASE results.

As we report in the manuscript, the subset of participants that took part in both sessions (GEEPI and 3D GRASE acquisitions) was small (N=4), as the 3D GRASE acquisitions were intended as a control for vasculature-related confounds. This small sample size does not support further statistical analyses; yet, we believe it to be a valuable control dataset. We tested the same participants with both GE-EPI and 3D GRASE to assess reproducibility of our results independent of the MR sequence used, as these sequences are known to be affected by vasculature-confounds at different degrees. To assess reproducibility, we correlated the fMRI adaptation index measured between sequences, across scanning sessions.

As reported in Figure 3—figure supplement 2, we observed substantial (r> 0.45) and statistically significant correlations for middle and superficial layers, suggesting that the laminar specificity of the GE-EPI results was unlikely to be significantly confounded by vasculature related confounds. We have now clarified in the revised manuscript our approach in using the GRASE measurements as a test of reproducibility.

Reviewer #3:The motivation and framing is to characterize "circuit properties of adaptation" but I have two issues with this. First, is the inference that neural adaptation is occurring. Yes, the signal is smaller when stimuli are repeated vs. when they are not, and this is *consistent* with adaptation. But, the origin(s) of fMRI repetition effects is controversial. Without a secondary measure – e.g., psychophysical evidence of adaptation (e.g., reduced sensitivity, tilt aftereffect, etc) in the adapted vs. non-adapted conditions or a direct physiological measure (e.g., spikes) – all we can say is that the fMRI signal is reduced during repetition. What I think this paper does a great job of doing is the set up for an experiment that specifically examines adaptation – for example, is there a specific layer-response that best predicts psychophysical differences in adaptation?

We agree with the reviewer that the neural mechanisms underlying fMRI adaptation (i.e. decreased BOLD response due to stimulus repetition) remain controversial. Our study aimed to investigate the feedforward vs. feedback mechanisms that contribute to fMRI adaptation by testing the laminar-specificity of fMRI adaptation. Neurophysiological recordings are beyond the remit of our study; yet our results showing laminar specificity have the potential to inspire future neurophysiological studies to test feedforward vs. feedback mechanisms of neural adaptation across cortical depths.

Our experimental design aimed to ensure that the comparison of BOLD signals across conditions was not confounded by differences in task performance. To this end, participants engaged in an attentional control task rather than in a behavioural task that measures perceptual adaptation. Following the reviewer’s suggestion, we conducted an additional behavioural experiment on the same participants who participated in the fMRI experiment. We used a classic tilt-aftereffect task to measure perceptual adaptation to orientation. In brief, participants were presented with blocks of gratings of either the same (adaptation block), or different orientations (non-adaptation). Following the presentation of 21 gratings, a test stimulus with orientation near vertical was shown, and participants were asked to indicate whether the test was oriented clockwise or anti-clockwise compared to vertical. During stimulus presentation, participants were asked to perform the same RSVP task as in the fMRI experiment to ensure that attention was maintained at fixation. Performance in the tilt after effect task showed a perceptual bias, that is, a significant shift in the distribution of responses away from the adapted orientation for the adaptation compared to non-adaptation condition (t(11)=-3.197, *p*=0.0085).

Correlating perceptual adaptation index and fMRI adaptation showed a significant correlation for superficial V1 layers (r=0.640, p=0.046) but not middle (r=0.562, p=0.091) or deeper V1 layers (r=0.563, p=0.09). Further, correlating perceptual adaptation index with mean fMRI adaptation across visual areas (V1, V2, V3, V4) showed similar results; that is significant correlation for superficial (r=0.706, p=0.039) but not middle (r=0.562, p=0.091), nor deeper (r=0.567, p=0.087) layers. These results suggest that the laminar-specificity of fMRI adaptation in visual areas relates to a behaviour measure of perceptual adaption (i.e. perceptual bias away from the adapted orientation).

The second issue is the inferences that are made between depth and feedback/feedforward processing. Take, for example, a measured difference in superficial layers. I don't understand how it is possible to know whether a change in the BOLD signal in superficial layers is due to neurons in these layers being affected by within-area circuits (e.g., from known connections between middle-layers to superficial layers) or due to feedback-mediated effects (as feedback affects both superficial and deep layers). In light of this, it's difficult to parse the first paragraph of the Discussion. "First, visual adaptation is implemented by recurrent processing of signals in visual cortex, as indicated by fMRI adaptation (i.e. BOLD decrease due to stimulus repetition) across layers with stronger effects in superficial than middle and deeper layers." What does "recurrent processing" mean here? If it means "feedback" shouldn't deeper layers have stronger effects? Does it mean with-area processing (middle/input -> superficial/output). Overall, I find the report of layer-specific responses interesting but I cannot infer the level of "circuit properties" the authors' wish to ascribe to the effects.Along similar lines, as a way to refresh my memory of layers/connections in early visual cortex, I looked at this recent paper: "Anatomy and Physiology of Macaque Visual Cortical Areas V1, V2, and V5/MT: Bases for Biologically Realistic Models". It is difficult to map the relatively simple characterization presented in the current paper with the real complexity described in the Vanni et al. paper. As just one example, from Vanni et al., "FF connections from V1 to V5 arise from layers 4B (both blobs and interblobs) and 6 and target primarily L4 and less so L3 of V5." "FB projections from V5 to V1 terminate predominantly in layers 4B and 6 (Maunsell and Van Essen 1983b; Ungerleider and Desimone 1986b; Shipp et al. 1989), that is, the source layers of the V1-to-V5 FF projection." There is very little consistency between this characterization and what is depicted in Figure 1A (though, I understand these quotes are specifically about MT-V1 connections).

We thank the reviewer for this comment. We agree that stronger fMRI adaptation in superficial layers could be due to a) output from middle layers where significant fMRI adaptation was also observed, suggesting recurrent processing within V1, b) feedback from higher visual areas or IPS. Although we did observe significant fMRI adaptation in deeper layers, previous work has shown that synaptic input to superficial layers may result due to increase in feedback signals carried by neurons that have dendrites projecting to the superficial layers and their cell bodies in deeper layers (Larkum, 2013). Our functional connectivity analysis sheds more lights into these mechanisms. We observed significant connectivity across deeper layers between V1 and V2, suggesting a local circuit based on recurrent processing within these regions and local feedback between them. Further, we observed significant connectivity across deeper layers between V1 and IPS, suggesting long-range connectivity between occipital cortex involved in sensory processing and parietal regions involved in expectation.

We agree with the reviewer that the scale of UHF imaging is limited compared to anatomical or neurophysiology studies; as a result, it is not possible to achieve one-to-one mapping between fMRI cortical depths and cyto-architectonically defined. Our analyses of cortical depth and functional connectivity follows the framework presented in Figure 1, as adopted by several previous UHF imaging studies investigating feedforward vs. feedback processing using a range of tasks and stimuli (e.g. perceptual illusions, finger tapping, word reading, attention; Huber et al., 2017; Kok et al, 2016; Moerel et al, 2020; Sharoh et al., 2019). Future studies using similar paradigms (i.e. stimuli, tasks) across animals and humans would be necessary to investigate functional correspondence across anatomical scales. We have clarified and discussed this point further in the revised manuscript.

Reviewer #2:1) "while feedback occipito-parietal connectivity" should be "while feedback was enhanced for occipito-parietal connectivity".

We have revised the text; See also response to point 1 of reviewer #1.

2) Introduction second sentence: This claim is lacking citations

We have included the following references: Clifford, 2002; Kohn, 2007

3) Results final paragraph: why isn't IPS considered visual cortex? It's definition in the Wang et al. atlas is based on topographic representation of visual space.

The IPS regions we tested are located in the posterior parietal cortex. The IPS definition in Wang et al., 2015, is based on a memory-guided saccade task. This is consistent with previous work (Silver and Kastner, 2009) that reports topographic representation in posterior parietal cortex using memory-guided saccades and spatial attention tasks rather than the standard visual stimulation protocol (flickering checkerboard) used for mapping visual retinotopic areas. In the revised manuscript, we clarify that our choice of IPS as a region of interest was motivated by the role of IPS in processing a) visual information, b) expectation due to stimulus familiarity.

Reviewer #3:Introduction paragraph 1 and paragraph 2: plethora. Maybe don't use plethora twice.

Thank you-we have revised the text.

Results: It seems like all ROIs should be in Figure 3A and remove Supplementary figure 3. All ROIs are in Figure 3B and all are included in the ANOVA. It would just be easier to refer to a figure that included all ROIs if ROIs are discussed in the ANOVA.

We thank the reviewer for this suggestion. We have now modified Figure 3 to include results from all ROIs. Figure 3—figure supplement 1 and 2 shows data from the behavioural task and GRASE data.

"Post-hoc comparisons showed significantly decreased fMRI responses for adaptation across cortical layers (deeper: t(14)=-3.244, p=0.006; middle: 126 t(14)=-3.920, p=0.002; superficial: t(14)=-4.134, p=0.001)." What are the t-tests comparing? Is that just in V1?

These tests related to V1 data. We have now revised the text to include post-hoc comparisons (Bonferroni corrected) related to the two-way ANOVA on fMRI adaptation index with ROI (V1, V2, V3, and V4), and cortical depth (deeper, middle, superficial) factors.

Error bars: Unfortunately, the error bars make it look as if there are no effects. Visually, looking at Figure 3A, I'd say the responses are identical across layers. And, I'd conclude the same about all ROIs in Figure 3B. In fact, as I write this, it's difficult to reconcile the p-values in the text and what is presented in the figures. I'm guessing the issue is the repeated-measures nature of the analysis in which case between-subject error bars are misleading. You might consider:

*https://www.researchgate.net/profile/Denis_Cousineau/publication/49619408_Confidence_intervals_in_within-subject_designs_A_simpler_solution_to_Loftus_and_Masson's_method/links/54e4c9300cf22703d5bf6023.pdf*

We thank the reviewer for pointing this out and for the resourceful information. We have now implemented the suggested method for computing error bars. This has helped with consistency between figures and statistics.